# Tissue damage drives co-localization of NF-κB, Smad3, and Nrf2 to direct Rev-erb sensitive wound repair in mouse macrophages

Dawn Z Eichenfield[1,2†], Ty Dale Troutman[1†], Verena M Link[1,3†], Michael T Lam[1,2,4], Han Cho[5], David Gosselin[1], Nathanael J Spann[1], Hanna P Lesch[1], Jenhan Tao[1], Jun Muto[6], Richard L Gallo[6], Ronald M Evans[5], Christopher K Glass[1,4*]

[1]Department of Cellular and Molecular Medicine, University of California, San Diego, San Diego, United States; [2]Biomedical Sciences Graduate Program, University of California, San Diego, San Diego, United States; [3]Department II, Faculty of Biology, Ludwig-Maximilian Universität München, Planegg-Martinsried, Germany; [4]Department of Medicine, University of California, San Diego, San Diego, United States; [5]Salk Institute for Biological Sciences, La Jolla, United States; [6]Department of Dermatology, University of California, San Diego, San Diego, United States

*For correspondence: cglass@ucsd.edu

†These authors contributed equally to this work

Competing interests: The authors declare that no competing interests exist.

**Abstract** Although macrophages can be polarized to distinct phenotypes in vitro with individual ligands, in vivo they encounter multiple signals that control their varied functions in homeostasis, immunity, and disease. Here, we identify roles of Rev-erb nuclear receptors in regulating responses of mouse macrophages to complex tissue damage signals and wound repair. Rather than reinforcing a specific program of macrophage polarization, Rev-erbs repress subsets of genes that are activated by TLR ligands, IL4, TGFβ, and damage-associated molecular patterns (DAMPS). Unexpectedly, a complex damage signal promotes co-localization of NF-κB, Smad3, and Nrf2 at Rev-erb-sensitive enhancers and drives expression of genes characteristic of multiple polarization states in the same cells. Rev-erb-sensitive enhancers thereby integrate multiple damage-activated signaling pathways to promote a wound repair phenotype.

## Introduction

Macrophages reside in all tissues of the body and play key roles in homeostasis, immunity, and disease. As immune cells, macrophages serve as sentinels of infection and injury and are active participants in both innate and adaptive immune responses. Detection of pathogens and tissue damage is mediated by a diverse array of pattern recognition receptors for pathogen associated molecular patterns (PAMPs) and damage associated molecular patterns (DAMPs), exemplified by the toll-like receptors (TLRs). Ligation of TLRs initiates profound changes in gene expression that include induction of chemokines, cytokines, anti-microbial peptides, and other factors that contribute to the innate immune response and influence adaptive immunity (*Ostuni et al., 2013*; *Lawrence and Natoli, 2011*). This response has been extensively characterized in vitro by treating cultured macrophages with specific TLR ligands such as bacterial lipopolysaccharide (LPS), a potent activator of TLR4 (*Kaikkonen et al., 2013*; *Escoubet-Lozach et al., 2011*; *Raetz et al., 2006*). TLR4 ligation regulates gene expression through signal transduction pathways culminating in the activation of latent signal-dependent transcription factors, which include members of the nuclear factor kappa-light-

chain-enhancer of activated B cells (NF-κB), interferon regulatory factor (IRF), and activator protein 1 (AP-1) families (*Medzhitov and Horng, 2009*). In macrophages, these factors are primarily directed to macrophage-specific enhancers that are selected by macrophage lineage determining transcription factors, PU.1 and CCAAT-enhancer-binding proteins (C/EBPs) (*Heinz et al., 2010*). The macrophage activation phenotype resulting from selective treatment with LPS, or in some cases a combination of LPS and interferon γ (IFNγ), is referred to as M(LPS) or M(LPS+IFNγ) activation (*Murray et al., 2014*), and is considered vital for the host response to bacterial or viral infection.

Macrophages also play important roles in regulating the resolution phase of inflammation as well as the repair of tissue damage. These functions are controlled by complex microenvironmental pathways that include reductionist signals such as transforming growth factor β (TGFβ) and interleukin 4 (IL4). TGFβ is generally considered to be an inducer of a 'de-activated' macrophage or M(TGFβ) phenotype, although it also acts as a potent chemo-attractant for monocytes and can potentiate their transition into activated cells (*Li et al., 2006*). Macrophages respond to TGFβ in both an autocrine and paracrine manner. For example, phagocytosis of apoptotic cells results in increased macrophage-mediated secretion of TGFβ and subsequent inhibition of inflammatory cytokine production (*Li et al., 2006*). In addition to dampening inflammatory responses, secreted TGFβ plays key roles in accelerating wound healing and fibrosis (*Schuppan and Kim, 2013*). At the transcriptional level, TGFβ signal transduction pathways function primarily in a Mothers against decapentaplegic homolog (SMAD)-dependent manner through Smad2-, Smad3-, and Smad4-mediated activation, as well as Smad7-mediated inhibition (*Massagué, 2012*). Like other signal-dependent transcription factors, ligation of TGFβ receptors causes the localization of Smad3 to genomic loci containing lineage-determining transcription factors (*Mullen et al., 2011*).

Regulation of macrophage gene expression by IL4 plays roles in containment of parasitic infections and in homeostatic functions of adipose tissue. IL4 acts through the IL4 receptor to activate signal transducer and activator of transcription 6 (Stat6) (*Lefterova et al., 2010*), which positively regulates gene expression upon binding to recognition elements in promoters and enhancers of target genes (*Li et al., 2006*). IL4 signaling regulates genes that control tissue remodeling, phagocytosis, scavenging, and the arginase pathway. The macrophage activation phenotype resulting from selective treatment with IL4 is referred to as M(IL4) and is considered vital for the role of macrophages in wound repair (*Van Dyken and Locksley, 2013*).

While M(LPS) or M(LPS+IFNγ), M(TGFβ), and M(IL4) macrophage phenotypes are clearly distinct in vitro, they result from selective activation of specific signaling pathways by strongly polarizing ligands. In vivo, macrophages encounter diverse combinations of signals that can change over time in response to physiological or pathological processes such as tissue injury. Recent studies show that these combinations of signals can influence the transcriptional landscape of macrophages in an input-specific fashion (*Lavin et al., 2014*; *Ginhoux et al., 2015*; *Gosselin et al., 2014*). However, how complex signals are integrated at the level of transcription and how reductionist stimuli (LPS, TGFβ, and IL4) can be used as a framework to predict how combinations of transcriptional regulators coordinate immune and tissue repair activities in complex tissue microenvironments remain largely unknown.

The Rev-erb nuclear receptor family consists of two members, Rev-erbα (also known as nuclear receptor subfamily 1, group D, member 1, NR1D1) and Rev-erbβ (also known as nuclear receptor subfamily 1, group D, member 2, NR1D2) (Rev-erbs), that regulate the expression of genes involved in the control of circadian rhythm (*Preitner et al., 2002*; *Liu et al., 2008*; *Cho et al., 2012*), metabolism (*Raspé et al., 2002*; *Le Martelot et al., 2009*; *Feng et al., 2011*; *Solt et al., 2012*), and inflammation (*Fontaine et al., 2008*; *Gibbs et al., 2012*). Rev-erbs mediate transcriptional repression through recruitment of the nuclear co-repressor (NCoR) and histone de-acetylase 3 (HDAC3) complex (*Yin and Lazar, 2005*). Rev-erbs lack the carboxy-terminal (AF2) transactivation domain, which is required for recruitment of co-activators (*Durand et al., 1994*). Genome-wide location analysis of Rev-erbα and Rev-erbβ in macrophages revealed thousands of binding sites, the vast majority of which resided at macrophage-specific enhancer-like regions of the genome established by PU.1 and other macrophage lineage determining factors (*Lam et al., 2013*). Gain and loss of function experiments indicated that Rev-erbs function to suppress the activities of these enhancers by repressing enhancer-directed transcription. While these studies provided insights into the functional significance of enhancer transcription, the biological consequences of the actions of Rev-erbs at these distal regulatory elements were not explored.

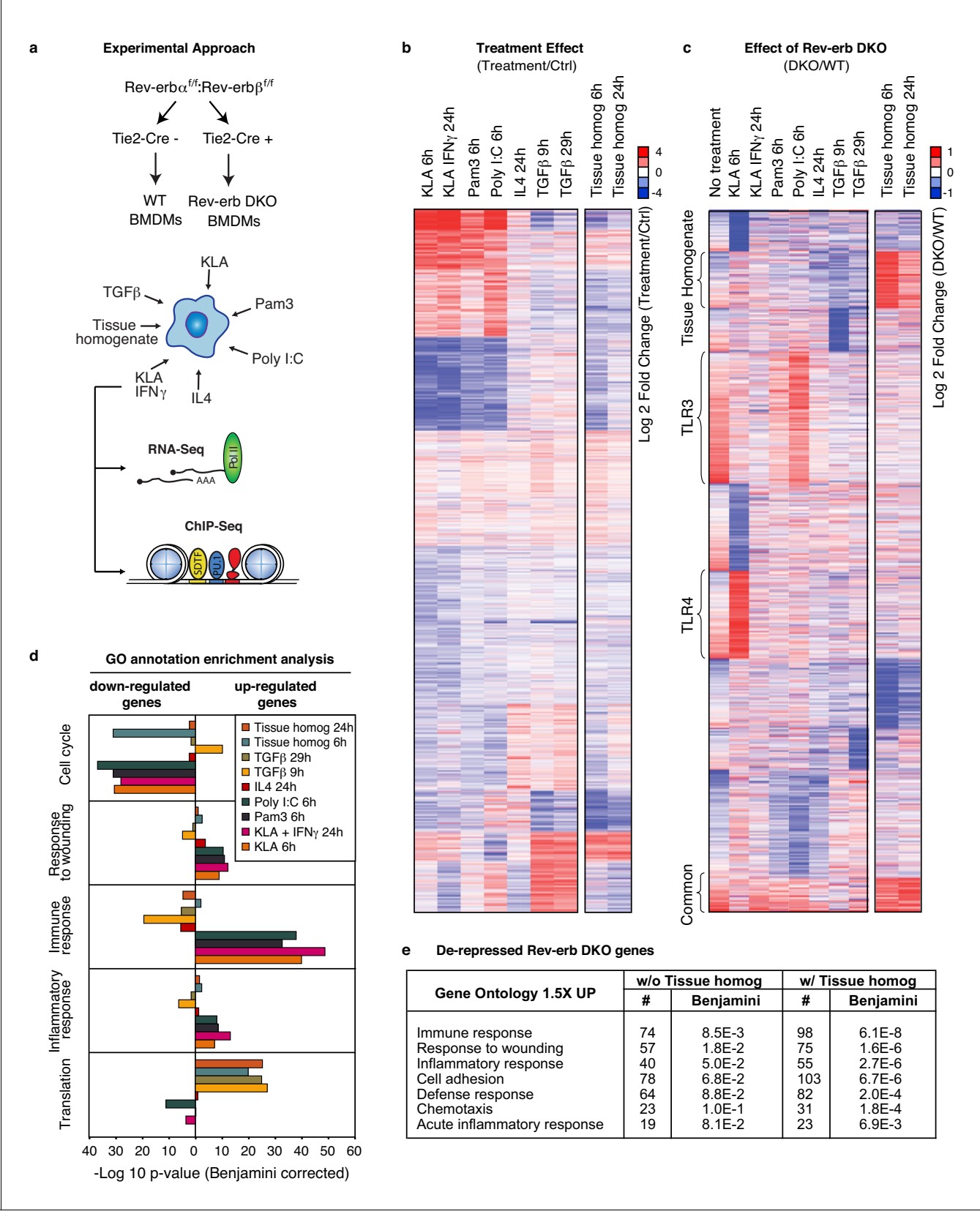

**Figure 1.** Overall impact of Rev-erb DKO on signal-dependent gene expression. (a) Schematic illustrating the experimental approach used in defining the global transcriptional program in WT and Rev-erb DKO bone marrow derived macrophages (BMDMs). (b) Heatmap showing genes captured by

*Figure 1 continued on next page*

*Figure 1 continued*

RNA-Seq associated with Rev-erb control after treatment with the indicated ligands compared to the basal state. Genes shown are those more than 1.5-fold differentially expressed in Rev-erb DKO macrophages compared to WT. Data is represented as log2 fold change between the basal state and treatment (untreated for 6 hr was used for comparison to KLA, Pam3, Poly I:C, TGFβ 9 hr, and tissue homogenate 6 hr; untreated for 24 hr was used for comparison to KLA + IFNγ, IL4, TGFβ 29 hr, and tissue homogenate 24 hr). Genes were clustered using k-means clustering (k = 10). For untreated samples, N = 4, for samples treated with Pam3, Poly I:C, KLA or KLA + IFNγ, tissue homogenate, or TGFβ, N = 3, and for samples treated with IL4, N = 2. The data for this heat map is accessible in *Figure 1—source data 1*. (c) Heatmap showing genes captured by RNA-Seq as differentially expressed 1.5-fold in the Rev-erb DKO macrophage compared to WT as indicated. Data is represented as log2 fold change between DKO and WT. Genes were clustered using k-means clustering (k = 10). For untreated samples, N = 4, for samples treated with Pam3, Poly I:C, KLA or KLA + IFNγ, tissue homogenate, or TGFβ, N = 3, and for samples treated with IL4, N = 2. The data for this heat map is accessible in *Figure 1—source data 2*. (d) Gene ontology analysis using David (*Huang et al., 2009a*, *2009b*) of genes shown in panel b. (e) Gene ontology analysis using David (*Huang et al., 2009a*, *2009b*) of genes demonstrating de-repressed expression in Rev-erb DKO macrophages by more than 1.5-fold in all of the conditions combined (w/o Tissue homog considers de-repressed genes in columns 1–8 of panel c (N = 2315), while w/ Tissue homog considers de-repressed genes in all columns of panel c (N=2614)).

The following source data and figure supplement are available for figure 1:

**Source data 1.** Source data for *Figure 1b* where each value represents the average normalized log2 fold change between the basal state and treatment state per column.

**Source data 2.** Source date for *Figure 1c* where each value represents the average normalized log2 fold change between the Rev-erb DKO macrophages compared to the WT per column.

**Figure supplement 1.** Rev-erb deletion strategy and efficiency.

Here, we provide evidence that Rev-erbs repress the transcription and function of signal-dependent enhancers that are targets of TLR, IL4, TGFβ, and DAMP signaling. Rather than exerting a pattern of repression that reinforces a particular polarization phenotype, Rev-erbs regulate subsets of signal responsive genes that span those associated with M(LPS) or M(LPS+IFNγ), M(TGFβ), and M(IL4) phenotypes, enriching for functions associated with wound repair. Consistent with these in vitro observations, deletion of Rev-erbs from the hematopoietic lineage in vivo results in accelerated wound repair. Unexpectedly, we found that a complex tissue injury signal directs genomic binding patterns for NF-κB p65 (p65), FBJ murine osteosarcoma viral oncogene homolog (Fos – a member of the activator protein 1, or AP-1, family), and Smad3 that differ substantially from those observed following selective treatments with a TLR4 agonist or TGFβ. In addition, by analyzing changes in enhancer signatures, we identified Nrf2 as an additional mediator of the transcriptional response to the tissue injury signal. While these transcription factors exhibit relatively little co-localization in response to single polarizing ligands, we observe substantial co-localization and enhancer activation in response to the complex tissue injury signal, resulting in transcriptional outcomes that are qualitatively different than the sum of single polarizing signals. These observations provide insights into how combinations of signals are integrated at a transcriptional level to result in context-specific patterns of gene expression.

## Results

### Rev-erb transcriptional activity varies according to polarizing signal

Our previous findings that Rev-erbs regulate transcription from signal-dependent enhancers (*Lam et al., 2013*) led us to investigate possible biological roles of Rev-erbs in influencing macrophage phenotypes (*Figure 1a*). To study the phenotypic contribution of Rev-erbs to signal-dependent gene expression in macrophages, we performed RNA-Sequencing (RNA-Seq) of poly(A) mRNA isolated from wild-type macrophages and those deficient for both Rev-erbα and Rev-erbβ (*Figure 1a*). Rev-erb double knockout (DKO) macrophages were generated from bone marrow differentiation of Tie2-Cre Rev-erbα$^{flox/flox}$ Rev-erbβ$^{flox/flox}$ (Rev-erb DKO) mice and compared to control macrophages derived from Cre-negative littermates (WT). Deletion of Rev-erbβ exons to generate a non-functional Rev-erbβ mRNA results in marked de-repression of Rev-erbα expression and increased expression of a DNA binding domain deleted form of Rev-erbα mRNA (*Sud et al.,*

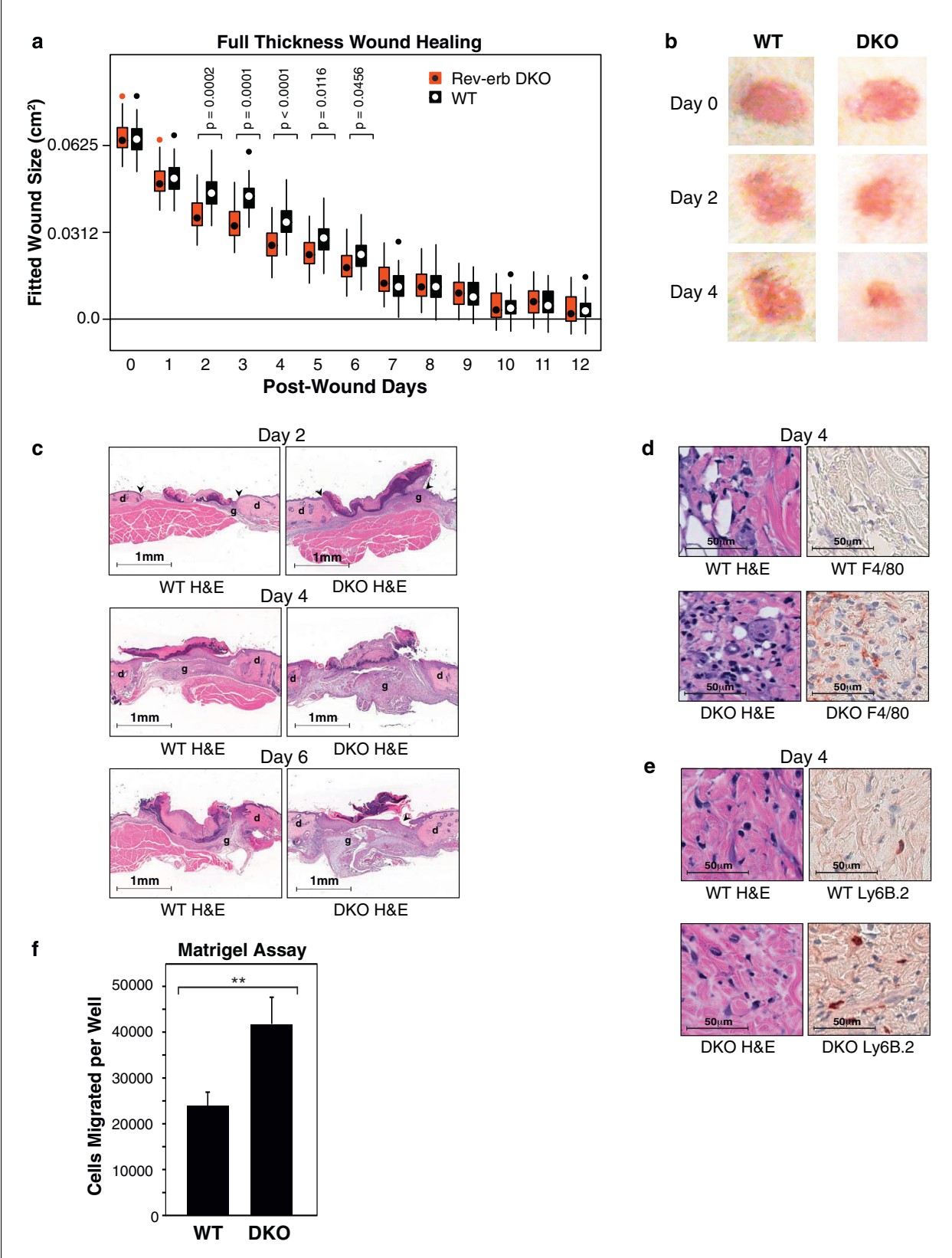

**Figure 2.** Rev-erb DKO bone marrow transplanted animals display enhanced wound closure in a full thickness wound healing model. (a) Wound size (cm$^2$) as fitted from a linear mixed effects model. Boxes denote the interquartile range and the median, whiskers denote the minimum and maximum

*Figure 2 continued on next page*

*Figure 2 continued*

values excluding outliers, and dots outside of the whiskers denote outlier observations. Data are pooled from three independent experiments as described in more detail in the Materials and methods. The p-values shown reflect comparisons with a p-value less than 0.05, as determined by the linear mixed effects model. (**b**) Macroscopic digital photographs of wound closure in WT and Rev-erb DKO bone marrow transplanted animals. (**c**) Histological images of wound healing in WT and Rev-erb DKO bone marrow transplanted animals taken at 2.5x magnification after 2, 4, and 6 days. Arrowheads show differential re-epithelialization between WT and Rev-erb DKO bone marrow transplanted animals. Abbreviations: g=granulation tissue, d=dermis. Images representative of two independent animals. (**d**) Day 4 hematoxylin and eosin (H&E), as well as F4/80 stained histological images taken at 20x magnification. Images representative of two independent animals. (**e**) Day 4 hematoxylin and eosin (H&E), as well as Ly6B.2 stained histological images taken at 20x magnification. Images representative of two independent animals. (**f**) Migration of WT and Rev-erb DKO macrophages through matrigel extracellular matrix for 24 hr (**p-value <0.01 two-tailed test, Data represent mean + SD from one of three experiments using 8 wells with cells pooled from 3 independent mice).

The following figure supplement is available for figure 2:

**Figure supplement 1.** Engraftment efficiency and quantification of circulating blood cells in WT and DKO chimeras.

*2007*) (*Figure 1—figure supplement 1a–b*). Similar effects can be seen following deletion of Rev-erbα exons (corresponding to the DNA-binding domain) with respect to Rev-erbβ de-repression (*Figure 1—figure supplement 1a–b*). Reduction of targeted Rev-erb exonic mRNA averaged 90% for Rev-erbα and 80% for Rev-erbβ (*Figure 1—figure supplement 1c*).

Activation of TLR3 with a synthetic double-stranded RNA analog, polyinosinic-polycytidylic acid (Poly I:C), TLR4 with Kdo2-lipid A (KLA), TLR1/2 with a synthetic triacylated lipopeptide, Pam3CSK4 (Pam3), and co-activation with KLA and IFNγ induced characteristic pro-inflammatory gene signatures (*Figure 1b and d*) in WT macrophages. In contrast, IL4 or TGFβ stimulation of macrophages resulted in the expected alternatively activated and de-activated gene profiles, respectively (*Figure 1b and d*).

Comparing the gene expression signature from WT and Rev-erb DKO macrophages, for the majority of genes, the magnitude of differential expression between WT and Rev-erb DKO macrophages varied depending on the polarization state (*Figure 1c*), in some cases only being observed under basal conditions, and in other cases only observed in response to a particular stimulus. These results suggest that the magnitude of differential expression in WT compared to Rev-erb DKO macrophages is highly dependent on polarization state.

## Rev-erb deficient animals display enhanced wound healing

Gene ontology analysis of mRNAs exhibiting differential expression (>1.5-fold de-repressed in DKO macrophages) in at least one of the single polarizing conditions revealed significant enrichment for genes involved in the response to wounding (*Figure 1e*). Notably, genetic loss of *Cx3cr1* and *Arg1* has been shown to hinder efficient wound healing in mice (*Campbell et al., 2013*; *Ishida et al., 2008*), suggesting that mice lacking Rev-erbs in cells of hematopoietic origin might exhibit more rapid wound healing. To test this hypothesis, we utilized a full thickness wound healing model (*Figure 2a*) in mice after bone marrow reconstitution with either WT or Rev-erb DKO bone marrow (*Figure 2—figure supplement 1a*). Bone marrow reconstitution efficiency exceeded 94% (*Figure 2—figure supplement 1b*). We found from three independent experiments that Rev-erb deficiency in bone marrow derived hematopoietic cells resulted in accelerated wound closure (*Figure 2a–b*). This was especially apparent on days 2–6 post-injury (*Figure 2a*), consistent with Rev-erb deficiency resulting in a faster response during the immune phase of wound healing.

Wounds from the Rev-erb DKO chimeric mice displayed greater immune cell infiltration and faster wound healing progression, characterized by enhanced re-epithelialization and increased granulation tissue development (*Figure 2c*), characteristics correlated with an accelerated immune response during wound healing. In addition, Rev-erb DKO bone marrow transplanted mice displayed more macrophages at the wound site on day 4 post-injury (*Figure 2d*), while neutrophil persistence at the wound site remained similar between WT and Rev-erb DKO transplanted mice (*Figure 2e*). Moreover, matrigel migration assays show increased extravasation of Rev-erb DKO macrophages when compared to their WT counterparts (*Figure 2f*). Flow cytometry analysis of circulating blood leukocytes from WT and Rev-erb DKO bone marrow transplanted animals (*Figure 2—figure supplement 1c–d*) showed no differences in the populations of Ly6C$^{low}$/Ly6C$^{high}$ circulating monocytes. These

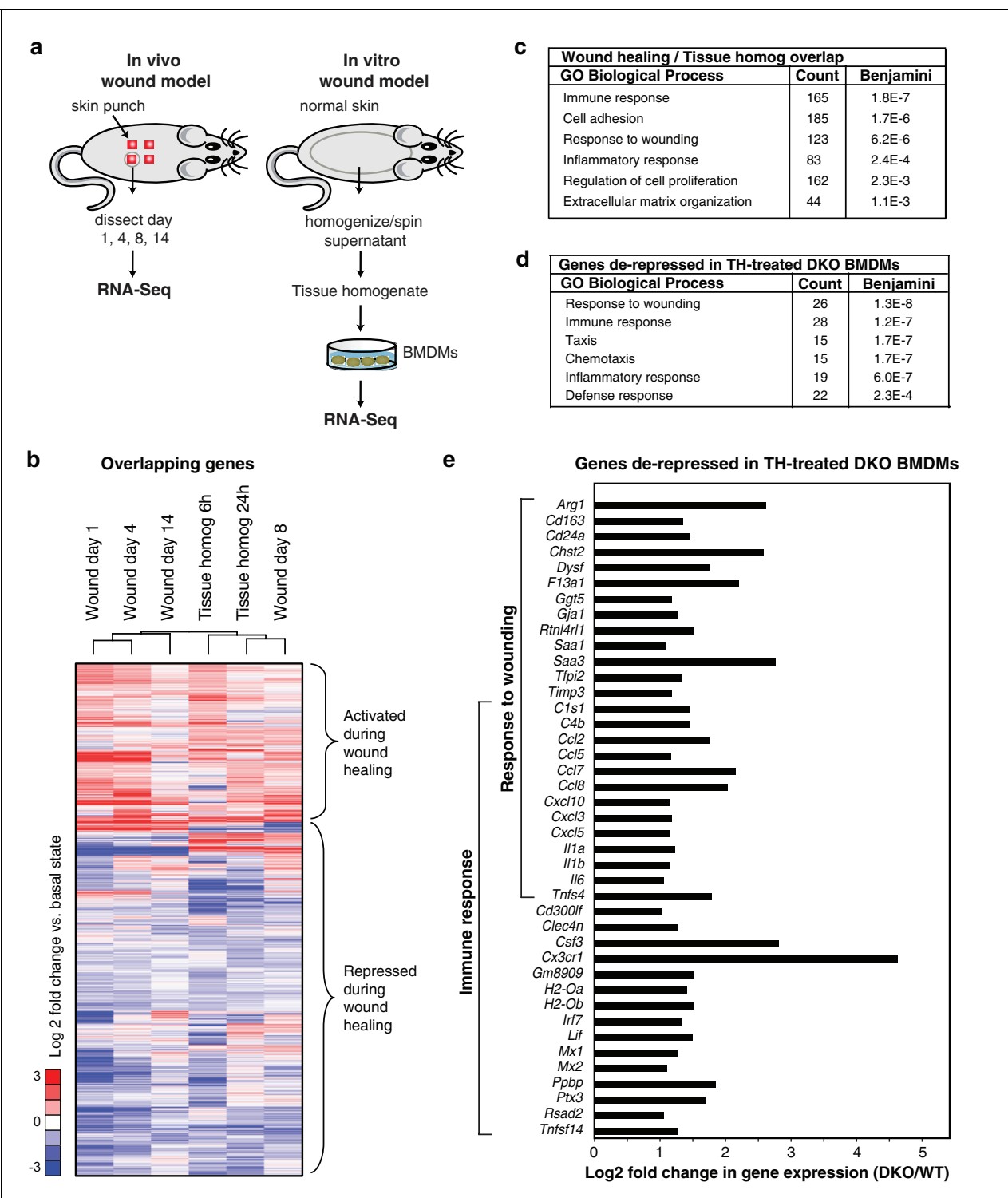

**Figure 3.** Rev-erb DKO macrophages display increased inflammatory responses to damaged tissue. (a) Schematic illustrating the experimental approach used comparing the transcriptional profile of in vivo wounds on day 1, 4, 8, or 14 post-wounding with macrophages treated in vitro with tissue homogenate after 6 or 24 hr. (b) Heatmap showing genes differentially expressed both in the in vivo mouse wound and in macrophages after in vitro stimulation with tissue homogenate (Tissue homog). Mouse wound genes from day 1, 4, 8, or 14 post-injury and macrophage tissue homogenate genes at 6 or 24 hr post-stimulation were compared to uninjured in vivo skin or unstimulated controls, respectively. Differentially expressed genes were those induced or repressed more than 1.5-fold compared to baseline. Genes were clustered using k-means clustering (k = 10). For unstimulated macrophages for 6 or 24 hr, N = 2, wound samples from day 1, 8, or 14, N = 2, macrophages stimulated with tissue homogenate for 6 or 24 hr, N = 3, and wound samples from day 0, or 4, N = 4. (c) Summary of gene ontology analysis using DAVID (*Huang et al., 2009a*, *2009b*) of overlapping wound healing and

*Figure 3 continued on next page*

*Figure 3 continued*

homogenate genes shown in **b** (N = 5590). *Figure 3—source data 1*. (d) Summary of gene ontology analysis using DAVID (*Huang et al., 2009a*, *2009b*) of genes de-repressed more than two-fold in Rev-erb DKO macrophages treated with tissue homogenate (TH) in comparison to WT macrophages (maximum de-repression after tissue homogenate treatment for 6 or 24 hr, N = 282). (e) Bar graphs depicting representative genes de-repressed more than two-fold (in log2 scale) in Rev-erb DKO macrophages after tissue injury (maximum de-repression after tissue homogenate treatment for 6 or 24 hr). Genes correspond to those associated with response to wounding and immune response categories in panel d. N as described in 3b.

The following source data is available for figure 3:

**Source data 1.** Source data for *Figure 3b* where each value represents the average normalized log2 fold change between the basal state and treatment state per column.

experiments suggest that the increased migration of macrophages into wounds may be cell autonomous changes in transcriptional output.

## Rev-erbs integrate macrophage responses to a complex wound signal

Classically, tissue injury of the skin, muscle, or organ systems induces an initial local inflammatory response, which is followed by subsequent regenerative processes involving macrophages and other immune cells, as well as mesenchymal stem cells (*Novak and Koh, 2013*). To devise an in vitro model of the acute in vivo response to wounding, we prepared a supernatant from homogenized skin (*Figure 3a*). This tissue homogenate (tissue homog/TH) provides a complex signal derived from components of disrupted cells (damage associated molecular patterns; DAMPs), the skin microbiome (microbial associated molecular patterns; MAMPs), and factors residing in the extracellular matrix (e.g., TGFβ). Tissue homogenate was used to stimulate WT and Rev-erb DKO macrophages for 6 and 24 hr, followed by RNA-Seq analysis. The gene expression signature of tissue homogenate-stimulated macrophages showed both similarities and differences when compared to the responses observed after treatment with TLR agonists, IL4, or TGFβ (*Figure 1b*).

In parallel, we performed temporal transcriptomic analysis of biopsied wounds during wound healing and compared them to unwounded skin (*Figure 3a*). Although myeloid cells represent only a small fraction of the total cells analyzed in the wound biopsy, 5590 genes exhibited concordant changes in expression with those observed following stimulation of macrophages with tissue homogenate (*Figure 3b*). Gene ontology analysis of this set of genes indicated significant enrichment for biological process terms related to response to wounding, immune response, and cell adhesion (*Figure 3c*). Response to wounding was the most highly enriched gene ontology term associated with genes de-repressed greater than two-fold in Rev-erb DKO tissue homogenate treated macrophages (282) followed by immune response and taxis (*Figure 3d*). De-repressed genes in Rev-erb DKO macrophages with gene ontology annotations linked to response to wounding and immune response are indicated in *Figure 3e*. These results indicate that tissue homogenate induces a Rev-erb-sensitive program of macrophage gene expression that substantially overlaps with the pattern of gene expression observed in response to wounding in vivo.

## Genes characteristic of alternate polarization states are co-expressed within individual cells

The approaches used thus far evaluated populations of cells. Genes associated with distinct polarization states resulting from activation with single ligands but exhibiting co-expression following treatment with tissue homogenate could reflect co-expression at the single cell level or mutually exclusive expression in subpopulations. To address this question, we performed RT-Q-PCR analysis of mRNA isolated from single cells maintained under control conditions or treated with tissue homogenate for 6 hr. We evaluated panels of mRNAs in triplicates corresponding to genes selectively activated by LPS or LPS+IFNγ, IL4, TGFβ, or tissue homogenate signals, as well as informative transcription factors and reference genes. After filtering for dead/duplicate cells and eliminating probes with altered melting curves, data was obtained for 30 genes in 80 control cells and 70 homogenate-treated cells. The distributions of expression values of genes in individual cells under control or tissue homogenate treatment conditions are illustrated in *Figure 4a*. Cells treated with

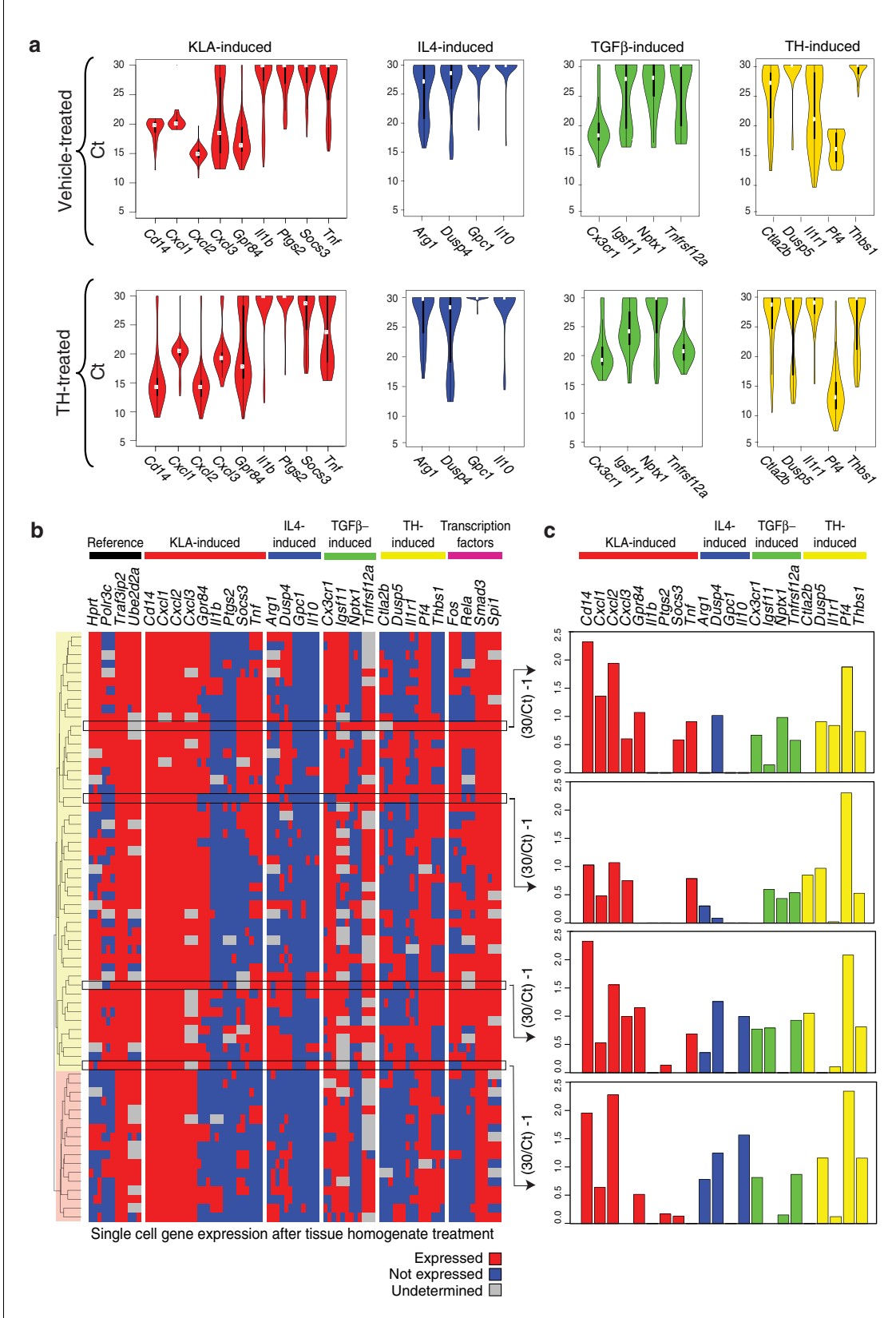

**Figure 4.** Genes characteristic of alternate polarization states are co-expressed within individual cells. (a) Violin plots of expression values for genes in the indicated categories as determined by single cell RT-Q-PCR from bone marrow derived macrophages treated for 6 hr with vehicle or tissue
*Figure 4 continued on next page*

*Figure 4 continued*

homogenate. Y-axis shows RT-Q-PCR CTs. Higher values indicate lower expression (30: gene product could not be detected). Values are averaged over 3 PCR replicates per gene. (**b**) Hierarchical clustering with Euclidean distance of single bone marrow derived macrophages treated with tissue homogenate based on expression (red) or lack of expression (blue) of the genes indicated at the top. Genes with alternating melting curves were treated as undefined (grey). PCR replicates are shown sequentially (N = 3). (**c**) RT-Q-PCR expression values for genes indicated above for four representative cells. Y-axis normalized to (30/CT) – 1. Higher values indicate higher expression (0: gene product could not be detected). Values are averaged over 3 PCR replicates per gene.

tissue homogenate were clustered in a binary fashion, according to whether the gene was expressed or not expressed. Notably, evaluating individual genes by column, a subset from each category of polarization states is expressed in the majority of cells (e.g., *Cxcl1, Dusp4, Cx3cr1, Pf4*) (*Figure 4b*). Conversely, evaluating the total set of genes across individual cells, genes from each polarization state can be expressed at similar levels in the same cell (*Figure 4c*). Of interest, clustering revealed two main groups that were distinguished by lack of detectable expression of *Fos* and *Rela*. Cells lacking *Fos* and *Rela* expression also exhibit reduced expression of subsets of genes in the M(LPS) or M(LPS+IFNγ), tissue homogenate, and transcription factor categories. Collectively, these findings indicate that while there is substantial heterogeneity in gene expression at the single cell level, genes characteristic of M(LPS) or M(LPS+IFNγ), M(TGFβ), and M(IL4) polarization states can be co-expressed in individual cells.

## Complex signals re-allocate transcription factors to novel genomic loci

To investigate mechanisms underlying effects of tissue homogenate on gene expression, we performed chromatin immunoprecipitation followed by high-throughput sequencing (ChIP-Seq) for histone 3 lysine 27 acetylation (H3K27ac), a histone modification associated with active enhancers and promoters (*Creyghton et al., 2010*), after 3 or 6 hr of control (Veh) or tissue homogenate stimulation. Treatment with tissue homogenate induced H3K27ac at ~2500 regions after 3 hr and ~5000 regions after 6 hr (*Figure 5a*). *De novo* motif analysis revealed binding sites for Nrf2, AP-1, and NF-κB motifs as among the most highly enriched sequences in these regions (*Figure 5b*).

Based on these motif findings, we initially performed ChIP-Seq analysis for p65 and Fos in macrophages treated with control or tissue homogenate. In addition, because SMAD motifs are difficult to retrieve using *de novo* motif analysis and tissue homogenate stimulation resembled treatment with TGFβ (*Figure 1b*), we performed corresponding ChIP-Seq analysis of Smad3. In each case, we observed that tissue homogenate induced a pattern of genomic binding sites that substantially differed from the pattern resulting from stimulation with the single ligands, KLA (p65 and Fos) or TGFβ (Smad3) (*Figure 5c*). These binding sites were also highly associated with tissue homogenate-induced gain of H3K27ac, consistent with their contribution to these changes in active chromatin (*Figure 5d*).

Examples of the binding patterns of Fos, p65, Smad3, and PU.1 in the vicinity of highly regulated genes are illustrated in *Figure 6a*, with responses of corresponding mRNAs to KLA, TGFβ, and tissue homogenate in WT and Rev-erb DKO macrophages shown in *Figure 6b*. Each genomic location contains numerous binding sites for each factor. PU.1 and Fos exhibit a high degree of constitutive binding, consistent with roles as pioneering factors that collaborate with each other and other macrophage lineage-determining factors, but also show quantitative changes in response to KLA and tissue homogenate. Smad3 and p65 both exhibit strong signal-dependent increases in ChIP-Seq signal at the majority of their binding sites. We note here that the starting conditions for KLA induction and tissue homogenate treatment differ, resulting in more constitutive binding of p65 in the vehicle control for tissue homogenate experiments.

Overall, there is a strong co-occurrence of p65 and Smad3 with pre-existing binding of Fos and PU.1, consistent with roles of these factors in establishing open regions of chromatin. Despite exhaustive efforts, we were not successful in determining high-confidence cistromes for endogenous Rev-erbs in BMDMs. We therefore considered the genomic locations of 7889 high-confidence binding sites occupied by both Rev-erbα and Rev-erbβ defined by ChIP-Seq of biotin-tagged proteins in RAW264.7 macrophages (*Lam et al., 2013*). For the de-repressed genes in the Rev-erb DKO, such as *Cx3cr1, Mmp9*, Arg1, and *Socs3* (*Figure 6a*), strong Rev-erb peaks coincide with at least one nearby enhancer-like region occupied by PU.1 and/or Fos, as well as p65 and/or Smad3 (e.g.,

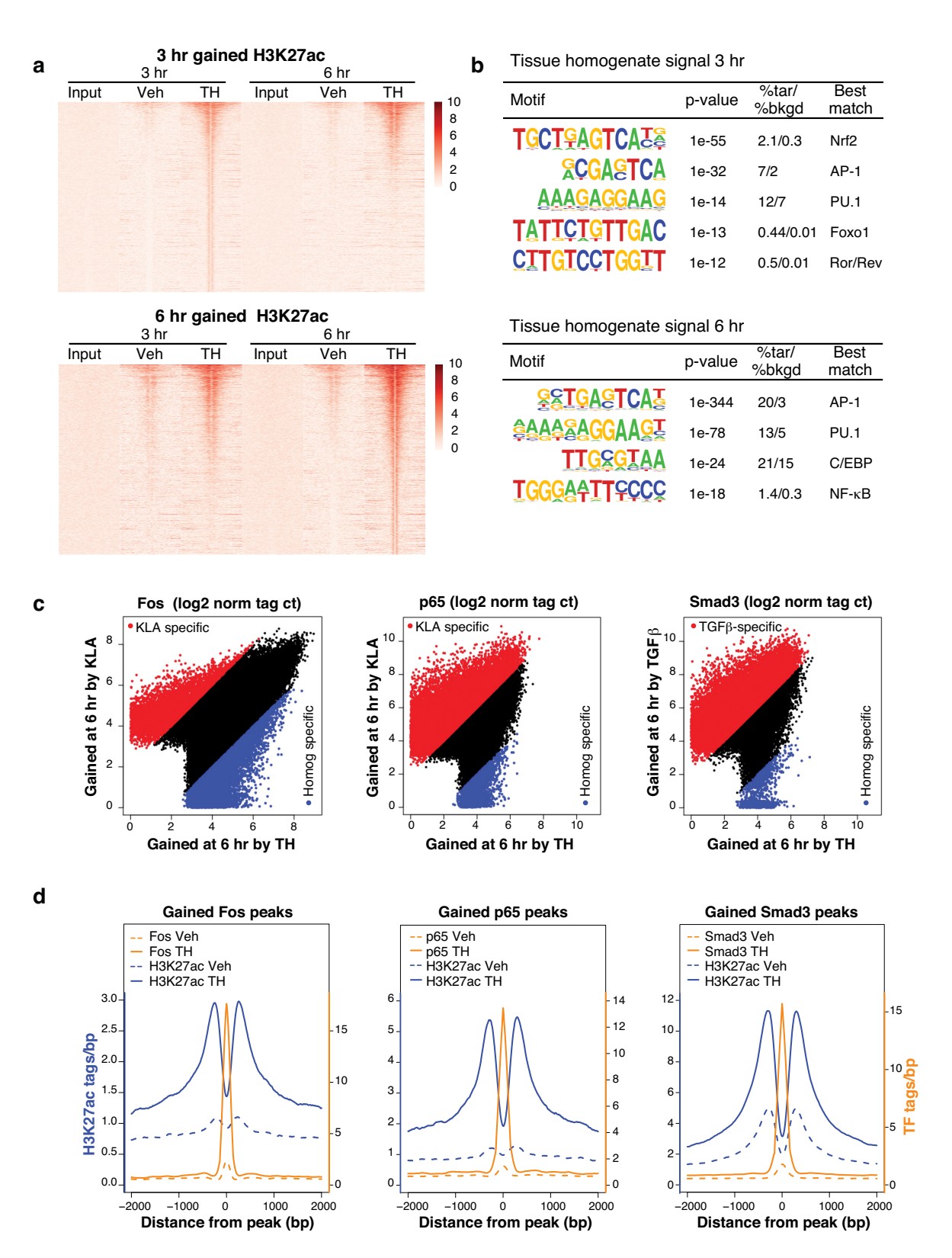

**Figure 5.** Complex transcriptional signals re-allocate transcription factors to novel genomic loci. (a) H3K27ac regions differentially gained upon treatment for 3 or 6 hr with tissue homogenate compared to treatment for 3 or 6 hr with the vehicle control. The heatmap shows a 6 kb window of
*Figure 5 continued on next page*

*Figure 5 continued*

normalized H3K27ac tag counts of the 2510 or 5005 homogenate gained regions at 3 or 6 hr, respectively, centered on the nucleosome free region (nfr). Input shows genomic background at these regions. N = 1. (**b**) Motifs enriched in the vicinity of gained H3K27ac sites after treatment with tissue homogenate for 3 or 6 hr using *de novo* motif enrichment analysis. (**c**) Comparison of Fos, p65, or Smad3 tag counts at genomic regions that contain Fos, p65, or Smad3 binding after stimulation with KLA, TGFβ, or tissue homogenate. Peaks found to be differentially gained (four-fold more tags) with KLA or TGFβ are colored red, while peaks found to be differentially gained (four-fold more tags) upon tissue homogenate treatment are colored blue. N = 1. (**d**) Quantification of H3K27ac, Fos, p65, and Smad3 ChIP-Seq tag counts in the 6 hr vehicle or tissue homogenate treated states centered on homogenate gained (using HOMER) Fos, p65, or Smad3 binding events. Dashed lines represent ChIP-Seq signal of the vehicle state and solid lines represent the signal after 6 hr of tissue homogenate stimulation. Blue represents H3K27ac signal, orange represents signal of the respective transcription factor. N = 1.

*Figure 6a*). Furthermore, the majority of Rev-erb binding sites identified in RAW264.7 macrophages co-localize with binding sites for their obligate co-repressor NCoR in BMDMs, strongly suggesting that Rev-erbs occupy a similar cistrome in these cells (*Figure 6a*).

Two observations were unexpected and noteworthy. First, a subset of enhancer-like regions occupied by Smad3 in cells treated with TGFβ were occupied by p65 in cells treated with KLA. Under conditions of stimulation with either KLA or TGFβ alone, the expectation is that these regions would be occupied by one factor, but not the other. However, in the context of tissue homogenate treatment, both factors are simultaneously bound (*Figure 6a*, yellow boxes). Second, and consistent with the results presented in *Figure 5c*, tissue homogenate treatment leads to binding sites for p65 and Smad3 that are not observed following treatment with KLA or TGFβ, respectively (*Figure 6a*, blue boxes). Furthermore, many of the new binding sites for p65 co-localize with Smad3 and vice versa. Consistent with these findings at individual genomic locations, motif analysis of tissue homogenate-specific SMAD binding sites (from *Figure 5c*) using TGFβ-specific SMAD sites as the background returned an NF-κB recognition motif as the second most highly enriched motif (*Figure 7a*).

To investigate whether co-localization of p65 and Smad3 in tissue homogenate-treated cells was a specific consequence of the complex signal, we performed ChIP-Seq analysis of p65 in macrophages selectively treated with TGFβ. We observed ~7400 p65 peaks, 5465 of which overlapped with the 39,825 peaks for Smad3 observed in TGFβ-treated cells, representing an overlap with 6% of the Smad3 peaks (*Figure 7b*). In contrast, we observed 20,858 p65 peaks and 13,975 Smad3 peaks in homogenate-treated cells, with p65 co-localizing with 11,379 (82%) of the Smad3 binding sites. Therefore, the complex tissue homogenate signal drives substantial co-localization of p65 with Smad3 that is not observed following selective treatment with TGFβ. These relationships are further illustrated for two representative genes, Arg1 and *Cxcl2*, in *Figure 7c*, in which yellow shading indicates regions where tissue homogenate induced p65 binding to regions occupied by Smad3 under either TGFβ or tissue homogenate treatment, whereas blue shading indicates regions in which both p65 and Smad3 binding are selectively observed following treatment with tissue homogenate.

## Nrf2 target genes and Nrf2 genomic binding are induced by tissue damage signals

Unexpectedly, the top enriched motif in tissue homogenate-specific SMAD sites is a binding site for NFE2L2, also known as Nrf2 (*Figure 7a*). This was also the top motif recovered from motif analysis of genomic regions exhibiting a gain in H3K27ac 3 hr following stimulation with tissue homogenate (*Figure 5b*). Nrf2 is a latent basic leucine zipper (bZIP) protein that is activated in response to cell injury and inflammation, and regulates the expression of antioxidant proteins that protect against oxidative damage (*Chen et al., 2015*). Evaluation of RNA-seq data from both the in vivo wound model and tissue homogenate-treated macrophages revealed upregulation of numerous Nrf2 target genes, including *Txn1, Sod2, Hmox1, Prdx6*, and *Nqo1*, suggesting that Nrf2 is activated in macrophages as part of the wound response (*Figure 7d*). We therefore performed ChIP-Seq analysis for Nrf2 in macrophages before and after tissue homogenate treatment. These experiments demonstrated that tissue homogenate increased the genome-wide binding of Nrf2 at thousands of genomic locations, a substantial fraction of which were observed to overlap with the tissue homogenate-induced binding sites for p65, Fos, and Smad3 (e.g., *Figure 6a*).

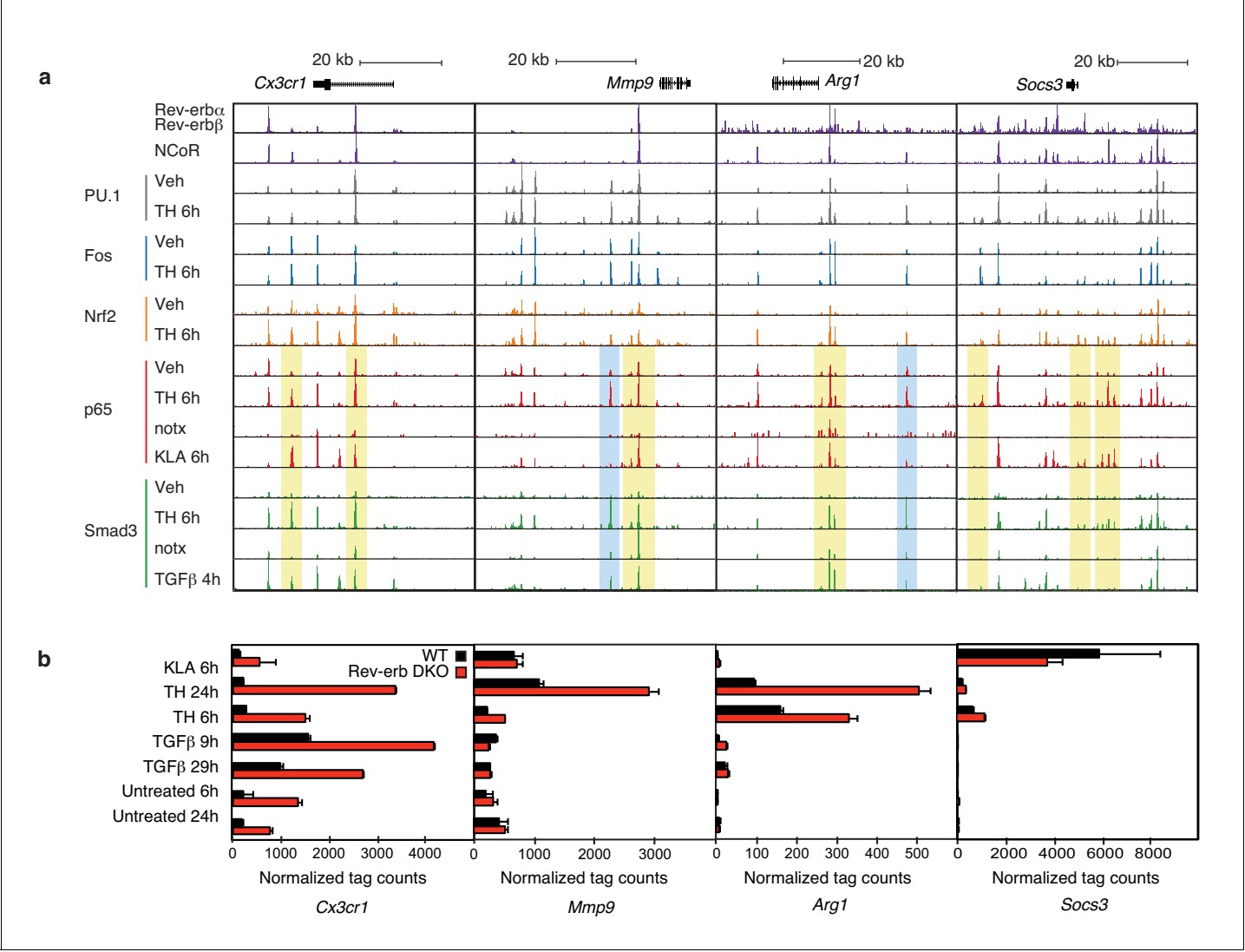

**Figure 6.** Locus-specific effects of Rev-erbs and signal-dependent transcription factors. (**a**) UCSC genome browser images depicting the genomic regions surrounding Rev-erb target genes, *Cx3cr1, Mmp9*, Arg1, or *Socs3*. The image shows the co-localization of Rev-erbα/Rev-erbβ, PU.1, Fos, Nrf2, p65, and Smad3 upon homogenate treatment. Yellow denotes gain of signal-dependent transcription factor peaks (p65 or Smad3) and Nrf2 after tissue homogenate stimulation that is not seen after treatment with one single polarizing signal. Blue denotes gain of signal-dependent transcription factor peaks (p65 and Smad3) after tissue homogenate stimulation that is not seen after treatment with either single polarizing signal. (**b**) Comparison of normalized RNA-Seq tag counts from WT or Rev-erb DKO macrophages stimulated as indicated. Error bars show standard deviation. For unstimulated 6 hr and 24 hr samples, N = 2, samples stimulated with KLA, tissue homogenate for 6 or 24 hr or TGFβ for 9 or 29 hr, N = 3.

## Tissue damage signals drive co-localization of PU.1, p65, Fos, Smad3, and Nrf2

To further explore the signal-dependent binding patterns of p65, Fos, Smad3, and Nrf2, we performed unbiased hierarchical clustering using peaks gained after stimulation with tissue homogenate or single stimuli. This analysis revealed that genomic occupancy of PU.1, Smad3, Nrf2, Fos, and p65 was most similar upon treatment of macrophages with the tissue homogenate signal, whereas patterns of transcription factor binding were more varied upon treatment of macrophages with individual stimuli (*Figure 7e*). This co-binding of transcription factors is further emphasized when comparing the overlap of the investigated transcription factors upon treatment of cells with tissue homogenate or the vehicle control (*Figure 7f and g*). This approach demonstrated co-localization of only 1.4% (893) of peaks in the vehicle state, which increased to 12.05% (7758) overlap when cells

were treated with tissue homogenate. This eight-fold increase in co-localization suggests that the combination of signals present in tissue homogenate induce co-binding of multiple transcription factors to enhancers that mediate the tissue injury response.

## NF-κB, Smad, Nrf2, and Rev-erb signaling pathways contribute to the integrated tissue damage response

Tissue homogenate contains a combination of DAMPs, MAMPs and other factors that have the potential to activate numerous signaling pathways. While ChIP-Seq experiments documented that tissue homogenate induces genomic binding of p65, Smad3, and Nrf2, these studies do not establish functional roles of these factors in the integrated transcriptional response. To address this question, we evaluated effects of chemical inhibitors of NF-κB, Smad3, and Nrf2 on gene expression in tissue homogenate-treated macrophages, using the IKK inhibitor VII to inhibit NF-κB activity, SB-43154 to inhibit TGFβ signaling, and glutathione to block the activation of Nrf2 (*Figure 8a–c*). These studies support the idea that each factor contributes to the integrated response to tissue homogenate. For example, activation of *Cx3cr1* by tissue homogenate was decreased upon targeting the NF-κB, TGFβ receptor, and Nrf2 pathways, supporting the involvement of all of these pathways in the regulation of this gene. Conversely, tissue homogenate activation of other genes was more dependent on specific signal-dependent pathways. For instance, *Dusp5* activation was sensitive to NF-κB inhibition (*Figure 8a*) while *Nptx1* activation was unaffected by NF-κB inhibition (*Figure 8a*). Surprisingly, *Socs3* activation was sensitive to both inhibition of NF-κB and TGFβ receptor signaling (*Figure 8a and b*) and *Ctla2b* was selectively sensitive to inhibition of TGFβ receptor signaling (*Figure 8b*). Established Nrf2 target genes *Txn1* and *Hmox1*, which were also induced by tissue homogenate, were repressed by glutathione co-treatment (*Figure 8c*). Finally, we investigated the ability of the Rev-erb agonist SR-9009 to influence the responses to tissue homogenate. This agonist repressed a subset of genes in tissue homogenate-treated macrophages, exemplified by *Cx3cr1*, *Gpr84*, and *Pgd* (*Figure 8d*). These results are consistent with these genes being de-repressed in Rev-Erb DKO macrophages.

## Discussion

Rev-erbs have been established to play general roles in the regulation of promoters of ubiquitously expressed genes such as *Bmal* that control the circadian rhythm (*Preitner et al., 2002*; *Liu et al., 2008*; *Cho et al., 2012*). However, the great majority of Rev-erb binding sites in macrophages are located at cell-specific enhancers, which are selected by macrophage lineage-determining factors such as PU.1 (*Lam et al., 2013*). These observations predicted that in addition to cell autonomous regulation of the circadian rhythm, Rev-erbs would also regulate a macrophage-specific program of gene expression. Here, using loss of function, transcriptomic, and epigenetic analyses, we demonstrate that Rev-erbs function to repress a network of genes associated with the response to wounding. Consistent with altered transcriptional responses observed in vitro, loss of Rev-erb expression in cells derived from the bone marrow compartment resulted in accelerated wound healing in the skin. As the Rev-erbs are deleted from all hematopoietic lineages in these experiments, further studies will be required to establish the relative contributions of macrophages and other bone marrow-derived cells to this phenotype. How this function of Rev-erbs might contribute to normal tissue homeostasis is as yet unclear. In vivo, Rev-erb expression is circadian (*Cho et al., 2012*), implying that the effects on macrophage gene expression observed in the present studies are likely to vary over the course of the day. Rev-erbs may thus act in a circadian manner to regulate aspects of tissue macrophage gene expression required for the normal turnover of extracellular matrix, tissue remodeling, and wound healing.

By evaluating the consequences of Rev-erb deficiency on macrophage gene expression in response to distinct polarizing signals in vitro, we found that the consequences of loss of function of Rev-erbs were dependent on the specific polarizing signal. Consistent with this, Rev-erbs co-localize with NF-κB p65 and AP-1 family member Fos at enhancers activated by TLR ligands, and with Smad3 at enhancers activated by TGFβ. Although of interest from a mechanistic standpoint, these findings are of uncertain relevance to functions of macrophages within tissue environments, which contain a multitude of signaling molecules that are sensed simultaneously. To model the complex environment associated with acute tissue damage, we treated macrophages with a supernatant of a

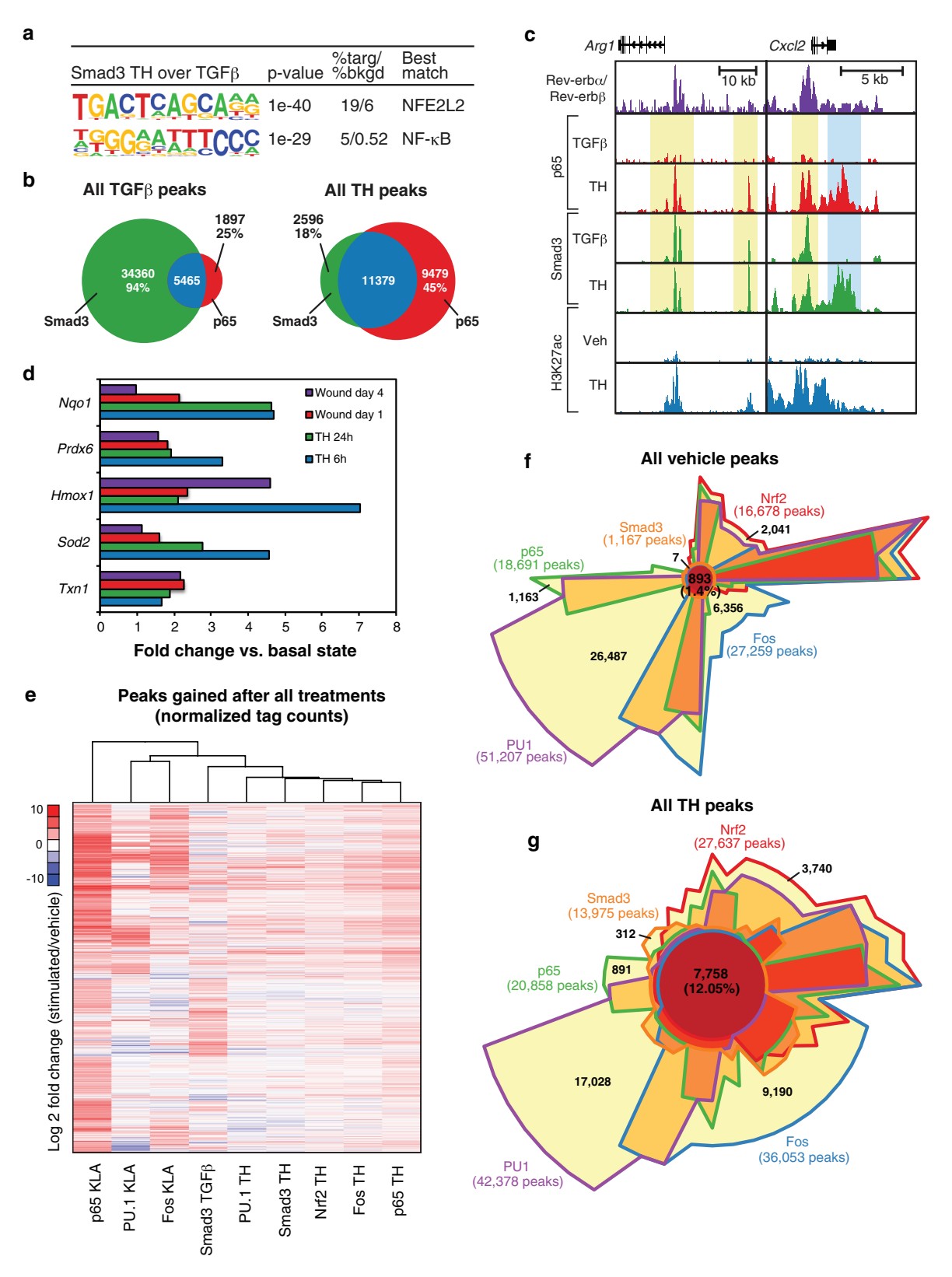

**Figure 7.** Signal-dependent transcription factors collaborate in response to complex stimuli. (a) Motifs enriched in the vicinity (200 bp) of Smad3 binding sites that are specific to tissue homogenate (induced four-fold), as compared to Smad3 binding sites that are specific to TGFβ (induced four-

*Figure 7 continued on next page*

*Figure 7 continued*

fold) using *de novo* motif enrichment analysis. (**b**) Venn diagrams depicting overlap of Smad3 and p65 after treatment with TGFβ (left) or tissue homogenate (right). Peaks have a minimal normalized tag count of 16. N = 1. (**c**) UCSC genome browser images depicting the genomic regions surrounding genes highly induced by tissue homogenate and not by TGFβ. Yellow denotes gain of signal-dependent transcription factor peaks (p65 or Smad3) after tissue homogenate stimulation that is not seen after treatment with one single polarizing signal. Blue denotes gain of signal-dependent transcription factor peaks (p65 and Smad3) after tissue homogenate stimulation that is not seen after treatment with either single polarizing signal. (**d**) Bar graphs depicting canonical Nrf2 genes induced during stimulation with tissue homogenate (green and blue) or during wound healing (purple and red). N as described in 3b. (**e**) Heatmap showing the log2 fold change of transcription factor tag counts at all genomic regions (minimum of 64 normalized tag counts in at least one condition per row) that are differentially gained (four-fold more tags) after stimulation as indicated. N = 1. (**f**) Chow-Ruskey Venn diagram depicting the overlap of all p65, PU.1, Fos, Smad3, and Nrf2 peaks after treatment with vehicle. Peaks have a minimal normalized tag count of 16. N = 1. (**f**) Same as (**g**) but depicting the overlap of all peaks after treatment with tissue homogenate. N = 1.

skin homogenate. While the specific identities and concentrations of the DAMPs, MAMPs, and other bioactive molecules in this homogenate are unknown, we provide evidence that the transcriptomic response of the macrophage to this mixture overlaps significantly with the transcriptional response observed in a skin wound, thereby validating its use. Through ChIP-Seq experiments, we demonstrate that this complex signal coordinately induces binding of NF-κB, AP-1, and Smad transcription factors. Furthermore, *de novo* motif analysis of activated enhancers led to the unexpected discovery that the tissue damage signal also acutely activates Nrf2. This finding illustrates the utility of enhancer analysis to identify transcriptional mediators of unknown environmental factors, providing a basis for subsequent directed analysis of corresponding upstream signaling pathways. Accordingly, the use of glutathione to neutralize reactive oxygen species, thus blocking the downstream disruption of the Kelch-like ECH-associated protein 1 (Keap1)-Cuilin 3 (Cul3) complex required for activation of Nrf2 (*Gorrini et al., 2013*; *Shibata et al., 2013*) provides evidence for its functional importance in the transcriptional response to the tissue damage signal. Similarly, the use of inhibitors of NF-κB and TGFβ provided corresponding support for functionally important roles of these transcription factors. Of course, there are likely to be many other signaling pathways and downstream transcription factors involved in the tissue damage response. Furthermore, Rev-erb deficiency likely modifies both basal and signal dependent transcriptional programs. To distinguish between 'prior' versus 'post-activation' roles of Rev-erbs in macrophages during wound healing may require the use of inducible Cre-expression strategies, as well as measurements of target gene expression in situ in macrophage infiltrated wounds.

Three additional observations are of particular interest. The first is that the complex signal provided by tissue homogenate induced co-expression of genes characteristic of distinct macrophage polarization states within individual cells. Second, we found that the tissue homogenate signal induced different genomic locations of p65, Fos, and Smad3 than were observed following KLA or TGFβ, respectively, resulting in co-binding at a large number of enhancer-like regions in the vicinity of tissue homogenate-induced genes. An important implication of these findings is that transcription factors binding maps are context-dependent and must be interpreted accordingly. We speculate that the observed co-localization of factors in response to the complex signal enables the appropriate integration of multiple relevant signaling components necessary for the initial acquisition of a wound repair phenotype (*Figure 8e*). Third, the present findings may have practical applications based on the development of small molecules that enhance or inhibit Rev-erb repressive activity (*Solt et al., 2012*). Delayed wound healing is observed in a number of pathological contexts, including in diabetics (*Falanga, 2005*; *Sen et al., 2009*) and in immunocompromised individuals (*Chen et al., 2013*; *Lin et al., 2011*). In these settings, it is possible that Rev-erb antagonists could be evaluated as a means of enhancing wound repair. Alternatively, a large number of devastating and largely untreatable diseases are characterized by exaggerated tissue fibrosis, such as idiopathic pulmonary fibrosis, interstitial renal fibrosis, and liver fibrosis (*Schuppan and Kim, 2013*). We demonstrate that a Rev-erb agonist can suppress a subset of genes that are de-repressed in the Rev-erb DKO and are regulated by the complex wound signal. Overall, our findings suggest that Rev-erbs act to repress a specific combination of genes downstream of multiple signaling pathways that collectively function in an integrated manner to promote the response to wounding (*Figure 8e*). It will therefore be of interest to evaluate whether defects in Rev-erb signaling are associated with these diseases and whether pharmacological modulation of Rev-erb might be of therapeutic benefit.

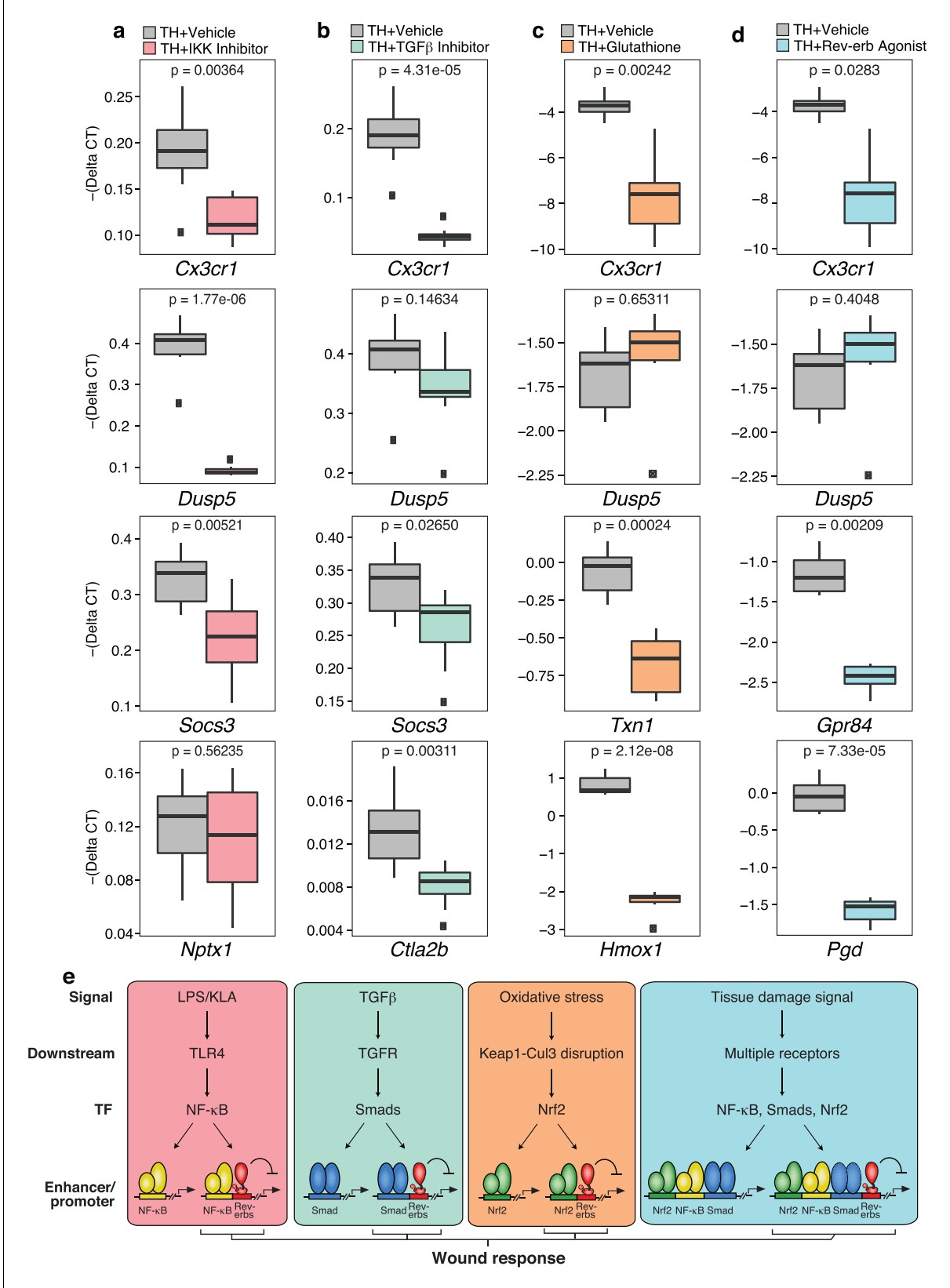

**Figure 8.** Chemical inhibition of multiple signal-dependent pathways results in decreased response to complex tissue homogenate signal. (**a**) Box and whisker plots of expression values for genes as determined by RT-Q-PCR from bone marrow derived macrophages treated with tissue homogenate for

*Figure 8 continued on next page*

*Figure 8 continued*

6 hr, and pre-treated for 1 hr with vehicle (gray) or 1 µM IKK inhibitor VII (pink). Y-axis shows RT-Q-PCR -(Delta CT), which is normalized to the housekeeping gene 36B4. Lower values indicate lower expression. N = 8 mice. P-values are shown comparing vehicle versus inhibitor treatment as determined by unpaired t-test. (b) Same as (a) but with pre-treatment for 1 hr with vehicle (gray) or 12.5 µM TGFβ inhibitor SB-43154 (green). N = 8 mice. (c) Same as (a) but samples were co-treated with tissue homogenate and vehicle (gray) or tissue homogenate and 15 mM glutathione (orange) for 6 hr. N = 6 mice. (d) Same as (a) but with pre-treatment for 1 hr with vehicle (gray) or 15 µM Rev-erb agonist SR-9009 (blue). N = 6 mice. (e) Working model showing that signal-specific stimuli (KLA/TGFβ/oxidative stress) activate their respective signal-dependent transcription factors NF-κB, Smads, and Nrf2, which bind to distinct sets of enhancers and promoters. Tissue damage signal activates all three factors simultaneously, which can co-occupy enhancers and promoters to generate a transcriptional response that is different than the sum of TGFβ, KLA, and oxidative stress mediated signaling. A subset of these sites that are co-bound and repressed by Rev-erbs are involved in regulating the macrophage response to wounding.

## Materials and methods

### Cell culture

Mouse bone marrow derived macrophages were obtained and cultured as previously described (*Heinz et al., 2010*). For cytokine stimulation studies, macrophages were cultured in RPMI-1640 (Invitrogen, Waltham, MA) supplemented with 16.7 ng/ml CSF1 (Shenandoah Biotechnology, Warwick, PA) and 0.5% heat-inactivated fetal bovine serum (FBS) (Hyclone, Logan, UT) overnight and then stimulated with Pam3CSK4 (300 ng/ml, InvivoGen, San Diego, CA), polyinosinic-polycytidylic acid (50 ng/ml, GE Healthcare Bioscience, Pittsburgh, PA), Kdo2-Lipid A (100 ng/ml, Avanti Polar Lipids, Alabaster, AL), recombinant interferon γ (10 U/ml, R&D Systems, Minneapolis, MN), interleukin 4 (20 ng/ml, R&D Systems), or tumor growth factor β (1 ng/ml, Cell Signaling, Danvers, MA) for the indicated time points. For ChIP-Seq experiments, cells treated with TGFβ or KLA were compared to untreated cells as a control.

For whole-skin tissue homogenate studies, skin from shaved wild type congenic mice was harvested and homogenized in RPMI-1640 supplemented with 0.5% heat-inactivated FBS using a Precellys 24 tissue homogenizer (6500 RPM, 4 × 20 s) and metal beads (2.8 mm beads, 2 mL tubes, Cayman Chemical, Ann Arbor, MI) according to the manufacturer's instructions. Skin homogenate was then centrifuged at 4000 RPM for 30 min at 4°C, and the supernatant filtered through a 0.2 µm filter (Nalgene, ThermoScientific, Rochester, NY). Approximately 50 ml of tissue homogenate was collected per mouse. To stimulate macrophages, macrophages were cultured in RPMI-1640 (Invitrogen) supplemented with 16.7 ng/ml CSF1 and 0.5% heat-inactivated FBS overnight. The following morning, the media was switched to either tissue homogenate or vehicle. 16.7 ng/ml CSF1 was added exogenously to both the homogenate and vehicle treatment conditions. For RNA-Seq replicates using BMDMs (where applicable), bone marrow of several mice were pooled and then cultured in different wells and processed independently.

For inhibitor experiments, macrophages were generated and cultured as described above, then pre-treated with 1 µM IKK inhibitor VII (Calbiochem, Billerica, MA), 12.5 µM TGFβ RI kinase inhibitor SB-43154 (Calbiochem), or 5 µM Rev-erb agonist SR-9009 (Burris laboratory) (*Lewis et al., 2013*) for 1 hr before treatment with tissue homogenate. For anti-oxidant experiments, macrophages were co-treated with tissue homogenate and 15 mM glutathione (Sigma, St. Louis, MO).

### Rev-erb DKO mice and genotyping

Rev-erbα and Rev-erbβ double floxed mice were generated as previously described (*Cho et al., 2012*) and crossed with Tie2-Cre (*Lam et al., 2013*). Breeding and genotyping were performed as previously described (*Lam et al., 2013*). Only males were used for wound healing experiments while both males and females were used for flow cytometry experiments to enumerate monocyte populations from peripheral blood. Littermates without the Tie2-Cre transgene were used as WT controls. All animal procedures were performed in accordance with the University of California, San Diego research guidelines for the care and use of laboratory animals (Permit Number: S01015).

### RNA isolation and RT-Q-PCR

Total RNA was harvested from tissue and cells using the RNeasy Mini Kit (Qiagen, Hilden, Germany) with in column DNase digestion performed according to the manufacturer's instructions. DNase-

treated RNA was used for cDNA synthesis using Superscript III (Invitrogen) according to the manufacturer's instructions.

For the IKK inhibitor VII and TGFβ inhibitor experiments, cDNA from biological replicates (N = 8) were assessed by quantitative polymerase chain reaction using SYBR GreenER Master Mix (Invitrogen) or SYBR Fast qPCR Master Mix (Kapa Biosystems, Wilmington, MA) on an Applied Biosystems 7300 Real-time PCR system or Step One Plus. For the glutathione and Rev-erb agonist experiments, cDNA from biological replicates (N = 6) were synthesized and assessed in technical triplicates by quantitative PCR using a Fluidigm Biomark HD (Fluidigm, San Francisco, CA). For statistical analysis, the delta CT was calculated for each biological replicate using 36B4 as the reference gene. Data were compared statistically using the t-test command in R.

## Single Cell RT-Q-PCR and analysis

BMDMs on petri plates were treated for 6 hr with vehicle or tissue homogenate in the presence of recombinant CSF-1. Following treatment, cells were removed by scraping and captured on a Fluidigm 17–25 micrometer C1 Single-Cell Auto Prep Array IFC or a 10–17 micrometer C1 Single-Cell Auto Prep Array IFC for homogenate or vehicle treated cells, respectively, according to the manufacturer's instructions. IFC positions having a single viable cell were noted and gene specific priming and pre-amplification was performed using the Fluidigm C1 instrument and the instrument protocol number 100–4904 H1. After cDNA synthesis, samples were harvested and stored at −20°C prior to detection of cDNA using Fluidigm 96.96 Dynamic arrays using the instrument protocol number 100–9792 A1. cDNA from individual cells was assessed in triplicate using the primers listed below.

For analysis of the data, melting curves of the triplicates were compared and samples with different melting curves or melting curves with more than one product were defined as NA. Gene products that could not be detected by Q-PCR were set to a CT of 30. A majority analysis was applied to the triplicates to calculate the average CT per primer pair per single cell (for violin plots [*Font-Vizcarra et al., 2012*]). Data was converted into binary data for gene expression heatmap, using 1 for expressed, 0 for not expressed (CT equals 30) and NA for undetermined. For cases where one sample of the triplicates had value 1, one had value 0 and one had value NA, NA was used as consensus. The heatmap for single cell analysis was created using hierarchical clustering with Euclidian distance in R.

## RT-Q-PCR Primers

| Gene target | Primer sequence |
| --- | --- |
| *36B4*-Forward | AGGGCGACCTGGAAGTCC |
| *36B4*-Reverse | CCCACAATGAAGCATTTTGGA |
| Arg1-Forward | TTTTAGGGTTACGGCCGGTG |
| Arg1-Reverse | CCTCGAGGCTGTCCTTTTGA |
| *Cd14*-Forward | CAGAGAACACCACCGCTGTA |
| *Cd14*-Reverse | CACGCTCCATGGTCGGTAGA |
| *Cd86*-Forward | CAGCACGGACTTGAACAACC |
| *Cd86*-Reverse | CTCCACGGAAACAGCATCTGA |
| *Ctla2b*-Forward | CTCATGCACCACTAGCCTCC |
| *Ctla2b*-Reverse | AGCAGGAAGACAGCACTGAA |
| *Cx3cr1*-Forward | CCATCTGCTCAGGACCTCAC |
| *Cx3cr1*-Reverse | CACCAGACCGAACGTGAAGA |
| *Cxcl1*-Forward | ACCCAAACCGAAGTCATAGCC |
| *Cxcl1*-Reverse | TTGTCAGAAGCCAGCGTTCA |
| *Cxcl2*-Forward | TGAACAAAGGCAAGGCTAACTG |
| *Cxcl2*-Reverse | CAGGTACGATCCAGGCTTCC |
| *Cxcl3*-Forward | ACCCAGACAGAAGTCATAGCCA |
| *Cxcl3*-Reverse | CTTCATCATGGTGAGGGGCT |

| | |
|---|---|
| *Dusp4*-Forward | CATCGAGTACATCGACGCAG |
| *Dusp4*-Reverse | ATGAAGCTGAAGTTGGGCGA |
| *Dusp5*-Forward | GCACCACCCACCTACACTAC |
| *Dusp5*-Reverse | CCTTCTTCCCTGACACAGTCAAT |
| *Fos*-Forward | TTTCAACGCCGACTACGAGG |
| *Fos*-Reverse | TCTGCGCAAAAGTCCTGTGT |
| *Gpc1*-Forward | GCCATGGAACTCCGGACC |
| *Gpc1*-Reverse | GCAGGTGCTCACCCGAGAT |
| *Gpr84*-Forward | AAACTGGGAACCTCAGTCTCCA |
| *Gpr84*-Reverse | GCCCAACACAGACTCATGGTA |
| *Hmox1*-Forward | GAGCAGAACCAGCCTGAACT |
| *Hmox1*-Reverse | AAATCCTGGGGCATGCTGTC |
| *Hprt*-Forward | GTTGGGCTTACCTCACTGCT |
| *Hprt*-Reverse | TCATCGCTAATCACGACGCT |
| *Igsf11*-Forward | GTGTCGCTGCTCGGTGT |
| *Igsf11*-Reverse | AGAATGACCTGTTCGGGCTG |
| *Il10*-Forward | GGTTGCCAAGCCTTATCGGA |
| *Il10*-Reverse | GGGGAGAAATCGATGACAGC |
| *Il1b*-Forward | TGCCACCTTTTGACAGTGATG |
| *Il1b*-Reverse | TGATGTGCTGCTGCGAGATT |
| *Il1r1*-Forward | GCTGACTTGAGGAGGCAGTT |
| *Il1r1*-Reverse | CATACGTCAATCTCCAGCGAC |
| *Nptx1*-Forward | TGGAGAACCTCGAGCAGTACA |
| *Nptx1*-Reverse | GTCAAGGCGCTCTCGATCTT |
| *Pf4*-Forward | CCCGAAGAAAGCGATGGAGAT |
| *Pf4*-Reverse | TTCAGGGTGGCTATGAGCTGG |
| *Pgd*-Forward | CTCCTCGACTCTGCTTCGTC |
| *Pgd*-Reverse | GCACAGACCACAAATCCATGA |
| *Polr3c*-Forward | TCTAAGAAGGGGCGATGGGA |
| *Polr3c*-Reverse | AGCCTCAGAACTCAGGGTCG |
| *Ptgs2*-Forward | AGCCAGGCAGCAAATCCTT |
| *Ptgs2*-Reverse | GGGTGGGCTTCAGCAGTAAT |
| *Rela*-Forward | CGGATTCCGGGCAGTGAC |
| *Rela*-Reverse | GAGGGGAAACAGATCGTCCA |
| *Smad3*-Forward | AAGAAGCTCAAGAAGACGGGG |
| *Smad3*-Reverse | CAGTGACCTGGGGGATGGTAAT |
| *Socs3*-Forward | TAGACTTCACGGCTGCCAAC |
| *Socs3*-Reverse | CGGGGAGCTAGTCCCGAA |
| *Spi1*-Forward | AAGCAGGGGATCTGACCAAC |
| *Spi1*-Reverse | AGTCATCCGATGGAGGGGC |
| *Thbs1*-Forward | GACAATTTTCAGGGGGTGCT |
| *Thbs1*-Reverse | AGAAGGACGTTGGTAGCTGAG |
| *Tnf*-Forward | GATCGGTCCCCAAAGGGATG |
| *Tnf*-Reverse | GTGGTTTGTGAGTGTGAGGGT |
| *Tnfrsf12a*-Forward | CAATCATGGCTTCGGCTTGG |
| *Tnfrsf12a*-Reverse | CTGCGGCGCCTGGTG |
| *Traf3ip2*-Forward | CCTGCTCCACCACTTACCTG |

| Traf3ip2-Reverse | TCTAGTTTCTAAGATCGCCACCG |
| Txn1-Forward | AGCCCTTCTTCCATTCCCTC |
| Txn1-Reverse | GGAAGGTCGGCATGCATTTG |
| Ube2d2a-Forward | AGCTGAGTGGGGCCTCG |
| Ube2d2a-Reverse | TCAATTCCTTGTGGATTCTCTTCA |

## RNA-Seq

Detailed protocols for RNA-Seq experiments have been previously described (*Kaikkonen et al., 2013*; *Heinz et al., 2013*). Briefly, total RNA was isolated using TRIzol LS (ThermoFisher Scientific) and resuspended with UltraPure water (ThermoFisher Scientific) supplemented with 1 μ/μL SUPER-ase-In (Ambion) then treated with TURBO DNA-free kit (Ambion). Poly(A) selection was performed using the MicroPoly(A)Purist kit (Invitrogen) according to the manufacturer's instructions. Poly(A) RNA was fragmented using RNA Fragmentation Reagents (Ambion) for 10 min at 70°C and purified by running through a Micro Bio-Spin P-30 column (Bio-Rad, Irvine, CA) according to the manufacturer's instructions. 30 ng RNA was utilized for subsequent library preparation.

For the following RNA samples: two replicates of the four day 0 in vivo wound samples, day 1 wound samples, day 4 wound samples, and day 14 wound samples, RNA library preparation was performed as previously described (*Kaikkonen et al., 2013*). Fragmented RNA was de-phosphory-lated using 1 μL T4 polynucleotide kinase (New England Biolabs, Ipswich, MA) and 5 μL 5x PNK buffer (0.5 M MES, 50 mM MgCl$_2$, 50 mM mercaptoethanol, 1.5 M NaCl, pH 5.5–5.8) supplemented with 1 μ/μL SUPERase-In for 45 min at 37°C, an additional 1 μL T4 polynucleotide kinase was added to the reaction, followed by incubation for 45 min, and subsequent heat-inactivation for 5 min at 70°C and ethanol precipitation overnight with glycogen. The pellet was resuspended in 5.5 μL nucle-ase free water supplemented with 1 μ/μL SUPERase-In and denatured for 5 min at 65°C. Poly(A)-tail-ing reaction was performed using 3.75 μ E. coli poly(A)-polymerase (New England Biolabs) in 10x poly(A)-polymerase buffer supplemented with ATP (50:1 molar ratio to RNA) and 1 μ/μL SUPERase-In for 30 min at 37°C. Reverse transcription was performed using Superscript III (Invitrogen). 8 μL RNA from the previous reaction, 1 μL 10 mM dNTP and 1 μL of the following oligo with custom barcodes (underlined and bolded): 5'-Phos **CA/TG/AC/GT**-GATCGTCGGACTGTAGAACTCT/idSp/ CAAGCAGAAGACGGCATACGATTTTTTTTTTTTTTTTTTTTTVN-3' were incubated for 3 min at 75°C and then chilled on ice. 1.7 μL 10x RT buffer, 3 μL 25 mM MgCl$_2$, 1.7 μL 0.1 M DTT, 0.5 μL SUPER-ase-In, and 0.9 μL Superscript III reverse transcriptase was added to the reverse transcription reac-tion and then incubated for 30 min at 48°C. After cDNA synthesis, 2 μL exonuclease I (New England Biolabs) was added to the reaction and incubated for 30 min at 37°C. The enzyme was inactivated and RNA hydrolyzed by adding 1 μl of 2 M NaOH and incubating for 20 min at 98°C. The reaction was then neutralized with 1 μl 2 M HCl. The cDNA was run on a 10% TBE-Urea gel (Invitrogen) and the gel was stained using SYBR gold (ThermoFisher Scientific). cDNA sized ~120–350 nucleotides were cut, gel purified, and precipitated overnight with ethanol and glycogen. Afterwards, cDNA was circularized by resuspending precipitated DNA in 10 μl circularization mix (7.5 μl of water, 1 μl 10x Reaction Buffer (Epicentre, Madison, WI), 0.5 μl of 1 mM ATP (final 0.05 mM), 0.5 μl of 50 mM MnCl$_2$ (final 2.5 mM), 0.5 μl CircLigase (100 μ/μl), (Epicentre)). Circularization was performed for 1 hr at 60°C, and the reaction was heat-inactivated for 15 min at 85°C. Circular single-stranded DNA was re-linearized by adding 3.3 μl of re-linearization mix (4x mix containing 100 mM KCl and 2 mM DTT) followed by 1 μl of APE 1 (15 μ; New England Biolabs). The reaction was incubated for 45 min at 37°C; an additional 1 μl APE 1 was added and the reaction was incubated for another 45 min. The enzyme was inactivated by incubating for 20 min at 65°C. The cDNA was amplified for 10–14 cycles using 0.1 μl Phusion polymerase (New England Biolabs), 2 μl 5x HF buffer, 0.2 μl 10 mM dNTP, 1 μl 5 M betaine, 4.7 μl water, and 0.5 μl of the following 10 μM primers: 5'-CAA GCA GAA GAC GGC ATA-3' and 5'-AAT GAT ACG GCG ACC ACC GAC AGG TTC AGA GTT CTA CAG TCC GACG-3'. The subsequent product was then gel purified from a 10% TBE gel (Invitrogen) using the ChIP DNA Clean & Concentrator Kit (Zymo Research Corporation, Irvine, CA).

For the following RNA-Seq samples: one replicate of no treatment 6 hr, one replicate of no treat-ment 24 hr, three replicates of polyinosinic-polycytidylic acid treatment 6 hr, two replicates of

Pam3CSK4 treatment 6 hr, one replicate of Kdo2-lipid A treatment 6 hr, one replicate of IL4 treatment 24 hr, and one replicate of Kdo2-lipid A and interferon-γ treatment 24 hr, strand-specific RNA sequencing libraries were prepared from poly(A) mRNA using a method similar to that previously described (Wang, 2011) with modifications described herein. Briefly, poly(A) enriched mRNA was fragmented, in 2x Superscript III first-strand buffer with 10 mM DTT (Invitrogen), by incubation at 94°C for 9 min, then immediately chilled on ice before the next step. The 10 μL of fragmented mRNA, 0.5 μL of random primer (Invitrogen), 0.5 μL of Oligo dT primer (Invitrogen), 0.5 μL of SUPERase-In (Ambion), 1 μL of dNTPs (10 mM), and 1 μL of DTT (10 mM) were heated at 50°C for three minutes. At the end of incubation, 5.8 μL of water, 1 μL of DTT (100 mM), 0.1 μL Actinomycin D (2 μg/μL), 0.2 μL of 1% Tween-20 (Sigma), and 0.2 μL of Superscript III (Invitrogen) were added and incubated in a PCR machine using the following conditions: 25°C for 10 min, 50°C for 50 min, and a 4°C hold. The product was then purified with RNAClean XP beads according to manufacturer's instructions and eluted with 10 μL nuclease-free water. The RNA/cDNA double-stranded hybrid was then added to 1.5 μL of Blue Buffer (Enzymatics, Beverly, MA), 1.1 μL of dUTP mix (10 mM dATP, 10 mM dCTP, 10 mM dGTP, and 20 mM dUTP), 0.2 μL of RNAse H (5 μ/μL), 1.05 μL of water, 1 μL of DNA polymerase I (Enzymatics), and 0.15 μL of 1% Tween-20. The mixture was incubated at 16°C for 1 hr. The resulting dUTP-marked dsDNA was purified using 28 μL of Sera-Mag Speedbeads (Thermo Fisher Scientific), diluted with 20% PEG8000, 2.5 M NaCl to final of 13% PEG, eluted with 40 μL EB buffer (10 mM Tris-HCl, pH 8.5), and frozen at −80°C. The purified dsDNA (40 μL) subsequently underwent end repair by blunting, poly(A)-tailing, and adapter ligation as described below.

All other RNA-Seq samples were prepared as described (Heinz et al., 2013). After RNA fragmentation and re-buffering with the Micro Bio-Spin P-30 column (Bio-Rad) according to the manufacturer's instructions, samples were resuspended with 16.5 μl of water. For de-capping using tobacco acid pyrophosphatase (TAP) (Epicentre), the following was added to the reaction: 2 μl 10x TAP buffer, 1 μl (20 μ) SUPERase-In (Ambion), 0.5 μl TAP; the reaction was then incubated for 2 hr at 37°C. Samples were then 3′ de-phosphorylated using T4 polynucleotide kinase (New England Biolabs); 0.5 μl 10x TAP buffer, 1.5 μl water, 0.5 μl 0.25 M MgCl$_2$, 0.5 μl 10 mM ATP, and 1 μL PNK was added to the reaction and incubated for 50 min at 37°C. After de-phosphorylation, samples were subsequently 5′ phosphorylated using T4 polynucleotide kinase in order to facilitate subsequent adapter ligation processes; 10 μL 10x T4 DNA ligase buffer, 63 μL water, and 2 μL PNK was added to the reaction and incubated for 60 min at 37°C. TRIzol LS was used to quench the reaction and extract phosphorylated RNA. RNA was resuspended in 4.5 μL water. For indexed library preparation, the 3′ adapter (0.5 μL 9 μM of a 5′-adenylated sRNA 3′ MPX adapter /5Phos/AG ATC GGA AGA GCA CAC GTC TGA /3AmMO/ (Integrated DNA Technologies, San Jose, CA)) was heat-denatured together with the RNA for 2 min at 70°C, placed on ice, and ligated with 100 U truncated T4 RNA ligase 2 (K227Q, New England Biolabs) in 10 μl 1x T4 RNA ligase buffer without ATP, containing 20 U/μL SUPERase-In and 15% PEG8000 for 2 hr at 16°C. Afterwards, 0.5 μL 10 μM MPX_RT primer 5′-GTG ACT GGA GTT CAG ACG TGT GCT CTT CCG ATC T-3′ (Integrated DNA Technologies, desalted) was added and annealed to the ligation product by incubating at 75°C for 2 min, then 37°C for 30 min, and then 25°C for 15 min. To ligate the 5′ adapter, 0.5 μl 5 μM hybrid DNA/RNA sRNA 5′h adapter 5′-GTT CAG AGT TCT ACA rGrUrC rCrGrA rCrGrA rUrC-3′ (Integrated DNA Technologies) was ligated to the 5′ end by adding 2 μl T4 RNA ligase buffer, 6 μl 50% PEG8000, 1 μl 10 mM ATP, 9.5 μL water, and 0.5 μl T4 RNA ligase 1 (New England Biolabs) for 90 min at 20°C. The reaction was then split in half (15 μl each) and 0.5 μL 10 μM MPX_RT primer was added to one 15 μL reaction. The reactions were incubated at 70°C for 1 min, then placed on ice. Reverse transcription was performed by adding 3 μL 10x RT buffer, 4.5 μL water, 1.5 μL 10 mM dNTP, 3 μL 0.1 M DTT, 1.5 μL RNaseOUT, and 1 μL Superscript III reverse transcriptase, then incubating for 30 min at 50°C. The cDNA was amplified for 10–14 cycles using 0.5 μL Phusion polymerase, 10 μL 5x HF buffer, 1 μL 10 mM dNTP, 5 μL 5 M betaine, and 0.25 μL of the following 100 μM primers: 5′-AAT GAT ACG GCG ACC ACC GAC AGG TTC AGA GTT CTA CAG TCC GAC G-3′ and TruSeq-compatible indexed primers (e.g. 5′-CAA GCA GAA GAC GGC ATA CGA GAT iii iii GTG ACT GGA GTT CAG ACG TGT GCT CTT-3′ (desalted, Integrated DNA Technologies, i signifies index nucleotides)). The subsequent product was then size selected for 175–225 base pair product and gel purified from a 10% TBE gel (Invitrogen) using the ChIP DNA Clean & Concentrator Kit. Libraries were PCR-amplified for 9–14 cycles, size selected by gel extraction, and quantified using the Qubit dsDNA HS Assay Kit (Thermo Fisher Scientific).

## ChIP-Seq

Previously published Rev-erbα and Rev-erbβ ChIP-Seq, and NCoR ChIP-Seq experiments, deposited as GSE45914 (*Lam et al., 2013*) and GSE27060 (*Barish et al., 2012*), respectively, were utilized for analyses. Detailed protocols for ChIP-Seq experiments have been previously described (*Kaikkonen et al., 2013*; *Heinz et al., 2010*, *Heinz et al., 2013*; *Li et al., 2013*). Antibodies against Fos (sc-7202), Nrf2 (sc-13032x), p65 (sc-372), and PU.1 (sc-352x) were purchased from Santa Cruz Biotechnology (Dallas, TX), against Smad3 (ab28379) from Abcam (Cambridge, UK), and against H3K27ac (39135) from Active Motif (Carlsbad, CA). Briefly, for Fos, Nrf2, p65, and Smad3 ChIPs, macrophages were first cross-linked in 2 mM dissuccinimidyl glutarate (Pierce 20593, Thermo Fischer) in PBS for 30 min, followed by subsequent 1% formaldehyde (Sigma) cross-linking in PBS for 10 min at room temperature. For H3K27ac and PU.1 ChIPs, cells were cross-linked using 1% formaldehyde in PBS for 10 min at room temperature. After cross-linking, glycine (Sigma) was added to a final concentration of 0.2625 M to quench the reaction. Subsequently, cross-linked macrophages were centrifuged (5 min, 1,200 RPM, 4°C), washed twice with PBS, and pellets were snap frozen and stored at −80°C. For ChIP of H3K27ac, p65, PU.1, Nrf2 or Smad3, frozen cell pellets were resuspended in cell lysis buffer (10 mM HEPES/KOH pH 7.9, 85 mM KCl, 1 mM EDTA, 1.0% IGEPAL CA-630 (Sigma), 1x protease inhibitor cocktail (Roche, Basel, Switzerland), 1 mM PMSF). After 5 min lysis on ice, cells were centrifuged (5 min, 4000 RPM, 4°C), and the supernatant was removed. The pellet was then resuspended in nuclear lysis buffer (10 mM Tris-HCl, pH 8.0, 100 mM NaCl, 1 mM EDTA, 0.5 mM EGTA, 0.1% Na-deoxycholate, 0.5% N-lauroylsarcosine, 1x protease inhibitor cocktail, and 1 mM PMSF) and the chromatin was sheared by sonication on wet ice with a Bioruptor Standard Sonicator (Diagenode, Denville, NJ) for three 15 min cycles each alternating 30 s on and 30 s off on the high setting. Additional Triton X-100 was added to the sonicated chromatin to 10% of the final volume and the lysate was cleared by centrifugation (5 min, 14,000 RPM, 4°C). Input was then saved for subsequent analysis.

For Fos ChIP, pellets were suspended in 50 mM Tris pH 8.0, 60 mM KCl, 0.5% IGEPAL, 1x protease inhibitor cocktail, and 1 mM PMSF, followed by 10 min of incubation on ice and centrifugation at 2000 ×g for 3 min at 4°C. The pellet was then suspended in 0.5% SDS, 10 mM EDTA, 0.5 mM EGTA, 50 mM Tris pH 8.0, 1x protease inhibitor cocktail, and 1 mM PMSF. The chromatin suspension was sheared by sonication on wet ice with a Bioruptor Standard Sonicator for three 15 min cycles each alternating 30 s on and 30 s off on the high setting, followed by centrifugation for 10 min at 15,000 RPM at 4°C. The chromatin was diluted 5x with 1% Triton X-100, 2 mM EDTA, 150 mM NaCl, 20 mM Tris pH 8.0, 1x protease inhibitor cocktail, and 1 mM PMSF. An input sample was saved for subsequent analysis.

Protein A or G Dynabeads (Invitrogen) pre-bound with antibody was added to the diluted cell lysate overnight at 4°C. Immunoprecipitated complexes were washed three times with 20 mM Tris/HCl pH 7.4150 mM NaCl, 0.1% SDS, 1% Triton X-100, 2 mM EDTA, three times with 10 mM Tris/HCl pH 7.4250 mM LiCl, 1% Triton X-100, 1% sodium deoxycholate, 1 mM EDTA, and two times with Tris-EDTA plus 0.1% Tween-20 before eluting two times with 50 µL elution buffer (TE, 1% SDS, 30 and 10 min, room temperature). Elution buffer was also added to the input. After pooling the eluted samples, the sodium concentration was adjusted to 300 mM and cross-links were reversed overnight at 65°C. Samples were treated with 0.5 mg/ml proteinase K for 1 hr at 55°C and 0.25 mg/ml RNase A for 1 hr at 37°C before DNA was isolated using the ChIP DNA Clean and Concentrator Kit according to the manufacturer's instructions. For library preparation, NEXTflex DNA barcode adaptors (BioO Scientific, Austin, TX) were ligated to the genomic DNA. Polymerase chain reaction mediated library amplification was performed and final libraries were size selected on 10% TBE gels (Invitrogen).

## High-throughput sequencing and data processing

RNA-Seq and ChIP-Seq libraries were sequenced for 50 cycles on an Illumina Hi-Seq 2000 (Illumina, San Diego, CA), sequenced for 51 cycles on an Illumina Hi-Seq 4000, or sequenced for 51 cycles on an Illumina NextSeq 2500 according to the manufacturer's instructions. ChIP-Seq reads were mapped to the mouse NCBI37/mm9 (*Ferreyra Garrott et al., 2013*) assembly using Bowtie (*Langmead et al., 2009*), allowing up to two mismatches. RNA-Seq reads were mapped to the mouse NCBI37/mm9 (*Ferreyra Garrott et al., 2013*) assembly using Tophat (*Trapnell et al., 2009*). Mapped reads were visualized using the UCSC genome browser (*Kent et al., 2002*) and

downstream data processing was performed using HOMER (*Heinz et al., 2010*), and R (*García-Oltra et al., 2013*).

## Genome-wide gene expression analysis with RNA-Seq

RNA-Seq analysis of genome-wide gene expression was performed using HOMER (*Heinz et al., 2010*). Differential expression was defined by a fold-change of at least 1.5-fold averaging over replicated datasets. For heatmap analysis, genes were clustered using k-means clustering (k = 10) in R. Gene ontology analysis was performed using DAVID Bioinformatics Resources 6.7 (*Huang et al., 2009a*, *2009b*).

## ChIP-Seq analysis

Genomic histone acetylation regions and transcription factor peaks were determined with HOMER using the findPeaks command default parameters of four-fold enrichment over the input, four-fold enrichment over local background, and normalization to 10 million mapped reads. For transcription factors, peaks were called using the '–style factor' parameter while histone acetylation regions were called using the '–style region' parameter. Histone regions were centered on nucleosome free regions using the '–nfr' parameter. For comparisons, called peaks from different data sets were merged using the mergePeaks command. Merging of transcription factor peaks or histone regions was done using the parameter '–size given'. To obtain differentially bound peaks/regions, tags were quantified from two data sets using the getDifferentialPeaks command. Peaks/regions were called as differentially gained if they had a four-fold enrichment of tag counts over the untreated/vehicle condition and a cumulative Poisson p-value less than 0.001. For heatmap analysis, peaks were clustered using hierarchical clustering in R.

## Bone marrow transplantation

Bone marrow harvested from WT and Rev-erb DKO mice was injected via the retro-orbital route into lethally irradiated (10 Gy) B6.SJL-Ptprc[a] Pepc[b]/BoyJ (CD45.1) (Jackson Lab, Sacramento, CA) or C57BL/6J (Harlem (now Envigo), Indianapolis, IN) 8 week old wild type congenic mice. Approximately 6–7 million bone marrow cells were injected per mouse. Transplanted mice were housed in autoclaved cages (changed every two days) and supplemented with antibiotics the day before irradiation until two weeks post-transplantation.

## Evaluating bone marrow transplant efficiency

To evaluate bone marrow transplant efficiency, whole blood from WT and Rev-erb DKO bone marrow transplanted mice was collected through cardiac puncture into EDTA tubes (Becton Dickinson, Franklin Lakes, NJ). 100 µL whole blood was washed once with PBS and resuspended in 2% FBS in PBS. Samples were blocked with 1 µL anti-mouse CD16/32 (eBioscience, San Diego, CA, 14-0161-82) for 15 min at room temperature. The following antibodies were utilized for staining: CD45 (Biolegend, San Diego, CA, 103122) and CD45.2 (Biolegend, 109813). Samples were incubated with directly labeled antibodies for 40 min (4°C in the dark). Stained cells were washed with 0.1% BSA in PBS, pelleted (1200 RPM, 5 min, 4°C), and lysed with hemolysin (Beckman Coulter, Brea, CA) for 20 s. Samples were quenched with 10x PBS, diluted, and gently washed before analysis using a LSR II flow cytometer (BD Bioscience, San Jose, CA). Unstained and single stains were used for setting up compensations and gating. Events were first gated on forward and side scatter to determine single events, before evaluation of other fluorescent markers.

## Monocyte enumeration from peripheral blood

Blood was collected from 16 chimeric mice per genotype into 0.5 ml K3 EDTA coated tubes. The volume of the blood was determined by pipetting and transferred to 5 ml round bottom tubes with 50 µl of Life Technologies (Carlsbad, CA) CountBright Absolute Counting Beads. Erythrocytes were lysed by addition of 4 ml eBioscience RBC lysis buffer with incubation at 4C for 5 mins. Cells were collected by centrifugation and the supernatant was carefully removed. Cells were washed once more and resuspended in buffer containing anti-CD16/CD32 (clone 93, BioLegend) and Zombie Aqua fixable viability dye (BioLegend). After 10 min, cells were stained with the following 2X antibody cocktail: anti-mouse CD11b BD Horizon BUV395 (clone M1/70, BD Biosciences), anti-mouse

CD19 BD Horizon BUV737 (clone1D3, BD Biosciences), anti-mouse CD115 Brilliant Violet 421 (clone AFS98, BioLegend), anti-mouse CD90.2 Brilliant Violet 785 (clone 30-H12, BioLegend), anti-mouse Ly6G FITC (clone 1A8, BioLegend), anti-mouse CD45.2 PE (clone 104, BioLegend), anti-mouse CD45.1 Alexa Fluor 647 (clone A20, BioLegend), and anti-mouse Ly6C APC/Cy7 (clone HK1.4, BioLegend). After 20 min, cells were washed, and counted on a Beckman Colter MoFlo Astrios EQ equipped with 355 nm, 405 nm, 488 nm, 561 nm and 640 nm lasers. Cells per µl were determined by following the manufacturer protocol for CountBright Absolute Counting Beads. Cells of interest were identified by excluding Zombie Aqua that fell within consecutive singlet gates using SSC and FSC. Donor derived monocytes were identified as CD45.2+, CD19-, CD90.2-, CD115+, Ly6G-, and CD11b+. Monocytes were further segregated based on expression of Ly6C. To test the dependence on hematopoietic derived expression of Rev-erb α/β on peripheral blood cell populations, we used a Welch two sample t-test using R.

## Wound healing studies

Wound healing studies were conducted 6–10 weeks post-transplantation. Briefly, 15.5 mL tert-amyl alcohol was added to 25 grams of 2,2,2 tribromoethanol (Sigma Aldrich Chemical) and dissolved overnight in a dark bottle to generate a stock solution. The subsequent solution was diluted with PBS, dissolved overnight, and filtered through a 0.2 µm filter to generate a working solution (20 mg/ml). To achieve anesthesia, 0.4–0.75 mg/g was administered intra-peritoneally. A 3 mm punch biopsy (Miltex, York, PA) was used to generate four wounds on the dorsal skin of each animal. Wounds were systematically photographed from a fixed distance daily. For macroscopic analysis, genotypes were blinded and the size of the wound was analyzed by Adobe Photoshop (San Jose, CA), and normalized to its size on Day 0. Mice were housed singularly throughout the duration of the study.

To assess the contribution of Rev-erb to wound healing, data from three independent experiments were combined and analyzed using a linear mixed effects model (*García-Gil et al., 2012*) using the R package 'nlme' (R script: wound model <- lme('wound size' ~ 'genotype' * 'time point', random=~1 | 'independent experiment' /'independent mouse' /'nested observation', data=data.file, na.action='na.exclude'). Genotype, time point, and their interaction, were modeled as fixed effects, whereas the observations at wound sites were treated as a random effect nested within the independent mouse, which in turn was treated as a random effect nested within the independent experiment, to account for the hierarchical nature of the study design. The numbers of biologically independent mice per time point are summarized in the below table.

| Day | WT chimera | Rev-erb DKO chimera |
|---|---|---|
| 0 | 29 | 28 |
| 1 | 28 | 27 |
| 2 | 26 | 25 |
| 3 | 17 | 17 |
| 4 | 21 | 20 |
| 5 | 22 | 20 |
| 6 | 22 | 20 |
| 7 | 13 | 12 |
| 8 | 20 | 16 |
| 9 | 12 | 9 |
| 10 | 7 | 7 |
| 11 | 12 | 9 |
| 12 | 7 | 7 |

## Histological analyses

At the indicated time points, mice were euthanized and wounds were harvested using a 6 mm punch biopsy (Miltex). Harvested wounds were cut along the mid-sagittal plane and paraffin-embedded.

Genotypes were blinded for subsequent histological analyses. The first section along the mid-sagittal plane was utilized for hematoxylin and eosin staining. Subsequent sections were utilized for immuno-histochemical analysis using the following primary antibodies: biotinylated anti-F4/80 (AbD Serotec (now Bio-Rad), MCA4978, 1:50 dilution), IgG (Dako, Glostrup Municipality, Denmark), rat anti-Ly6B.2 (AbD Serotec (now Bio-Rad), MCA771GA, 1:200), and the following secondary antibodies: biotinylated anti-rat (1:500, BD ), as well as HRP-conjugated streptavidin (1:500, Jackson Laboratory), Briefly, slides were de-paraffinized and washed three times in 0.1% Tween-20 PBS. Blocking was performed sequentially using 3% hydrogen peroxide (10 min), 1% BSA in 0.1% Tween-20 PBS (10 min), 0.1% avidin (10 min), and 0.01% biotin (10 min). Three washes were performed between each blocking step using 0.1% Tween-20 PBS. Antigen retrieval was performed using proteinase K (Dako, S3020), followed by three washes and subsequent overnight incubation with the indicated primary antibodies. After three washes, slides were incubated with the indicated secondary antibodies for 30 min and developed using AEC Peroxidase Substrate Kit (Vector Labs, Burlingame, CA, SK-4200) according to the manufacturer's instructions. Counterstaining was performed using Mayer's Hematoxylin (Sigma, MHS16), after which samples were mounted in an aqueous gel mount (Vecta-mount, Vector Labs, H-5501).

## In vitro matrigel migration assays

In vitro matrigel migration assays were performed as previously described (*Ogawa et al., 2004*). Briefly, macrophages were cultured in RPMI-1640 (Invitrogen) supplemented with 0.5% heat-inacti-vated FBS (Hyclone) for 24 hr and resuspended at a density of 1 million cells per milliliter. 100 μL of macrophages was added to the top chamber of a transwell (Corning, Corning, NY) while 650 μL of media was added to the bottom chamber. Macrophages were allowed to migrate through basement membrane extract (Corning, 3458) for 24 hr. Afterwards, the wells were briefly washed with PBS, and migrated macrophages were dissociated from the membrane and incubated with Calcein AM. Relative fluorescence was measured using a SpectraMax M3 plate reader (Molecular Devices, Sunny-vale, CA) and the SoftMax Pro software (485 nm excitation, 520 nm emission). A standard curve was used to convert relative fluorescence to cell numbers.

## Sequencing data

All sequencing data used in this manuscript has been submitted to GEO under the accession GSE72964. This data can be accessed by reviewers through the following link: http://www.ncbi.nlm.nih.gov/geo/query/acc.cgi?token=ejixaiswxlqnjiv&acc=GSE72964.

## Acknowledgements

We would like to thank Martina P Pasillas for assistance with flow cytometry and Liwen Deng at the UCSD Mouse Phenotyping Core for preparation of histological slides and immunohistochemical staining. We would also like to thank Lynn Bautista and Leslie Van Ael for their assistance with figure and manuscript preparation, as well as Dr. Thomas Burris for his kind gift of the SR-9009 Rev-erb ligand. These studies were supported by NIH Grants DK091183, DK063491, CA173903, and GM085764. DZE was supported by NIH training grant (T32-GM007198), and an American Heart Association Predoctoral Fellowship (12PRE11610007). TDT was supported by the National Cancer Institute of the National Institutes of Health under Award Number T32CA009523. HPL was supported by the Finnish Cultural Foundation, the North Savo Regional Fund, the Finnish Foundation for Cardiovascular Research, and the Maud Kuistila Memorial Foundation.

## Additional information

### Funding

| Funder | Grant reference number | Author |
| --- | --- | --- |
| National Institutes of Health | T32-GM007198 | Dawn Z Eichenfield |
| American Heart Association | 12PRE11610007 | Dawn Z Eichenfield |
| National Cancer Institute | T32CA009523 | Ty Dale Troutman |

| | | |
|---|---|---|
| Suomen Kulttuurirahasto | North Savo Regional Fund | Hanna P Lesch |
| Pohjois-Savon Rahasto | | Hanna P Lesch |
| Maud Kuistilan Muistosäätiö | | Hanna P Lesch |
| Suomen Kulttuurirahasto | The Finnish Foundation for Cardiovascular Research | Hanna P Lesch |
| National Institutes of Health | DK091183 | Christopher K Glass |
| National Institutes of Health | DK063491 | Christopher K Glass |
| National Institutes of Health | CA173903 | Christopher K Glass |
| National Institutes of Health | GM085764 | Christopher K Glass |

The funders had no role in study design, data collection and interpretation, or the decision to submit the work for publication.

## Author contributions

DZE, TDT, VML, Designed the study, Performed experiments, Analyzed and interpreted the data, Wrote the manuscript; MTL, Performed experiments, Analyzed and interpreted the data; HC, Performed experiments, Provided essential mouse models; DG, Performed experiments; NJS, Performed experiments, Developed RNA-Seq methods, Analyzed and interpreted the data; HPL, Designed the study, Performed experiments; JT, JM, Analyzed and interpreted the data; RLG, Designed the study, Analyzed and interpreted the data; RME, Analyzed and interpreted the data, Provided essential mouse models; CKG, Designed the study, Analyzed and interpreted the data, Wrote the manuscript

## Author ORCIDs

Dawn Z Eichenfield, http://orcid.org/0000-0002-1511-3804
Ty Dale Troutman, http://orcid.org/0000-0001-8925-8080
Verena M Link, http://orcid.org/0000-0002-3207-312X
Christopher K Glass, http://orcid.org/0000-0003-4344-3592

## Ethics

Animal experimentation: All animal procedures were in accordance with the University of California, San Diego research guidelines for the care and use of laboratory animals (Permit Number: S01015).

# Additional files

## Major datasets

The following dataset was generated:

| Author(s) | Year | Dataset title | Dataset URL | Database, license, and accessibility information |
|---|---|---|---|---|
| Eichenfield DZ, Troutman TD, Link VM, Lam MT, Cho H, Gosselin D, Spann NJ, Lesch HP, Tao J, Muto J, Gallo RL, Evans RM, Glass CK | 2016 | Tissue damage signals drive co-localization of NF-$\kappa$B, Smad3, and Nrf2 to direct a Rev-erb sensitive wound repair program in macrophages | http://www.ncbi.nlm.nih.gov/geo/query/acc.cgi?acc=GSE72964 | Publicly available at the NCBI Gene Expression Omnibus (Accession no: GSE72964) |

The following previously published datasets were used:

| Author(s) | Year | Dataset title | Dataset URL | Database, license, and accessibility information |
|---|---|---|---|---|
| Lam MT, Cho H, Lesch HP, Gosselin D, Heinz S, Tanaka-Oishi Y, Benner C, Kaikkonen MU, Kim AS, Kosaka M, Lee CY, Watt A, Grossman TR, Rosenfeld MG, Evans RM, Glass CK | 2013 | Rev-Erbs repress macrophage gene expression by inhibiting enhancer-directed transcription | http://www.ncbi.nlm.nih.gov/geo/query/acc.cgi?acc=GSE45914 | Publicly available at the NCBI Gene Expression Omnibus (Accession no: GSE45914) |
| Barish GD, Yu RT, Karunasiri MS, Becerra D, Kim J, Tseng TW, Tai LJ, Leblanc M, Diehl C, Cerchietti L, Miller YI, Witztum JL, Melnick AM, Dent AL, Tangirala RK, Evans RM | 2012 | A Bcl6-Smrt/Ncor repression program controls atherosclerosis | http://www.ncbi.nlm.nih.gov/geo/query/acc.cgi?acc=GSE27060 | Publicly available at the NCBI Gene Expression Omnibus (Accession no: GSE27060) |

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
