## [Decision Letter]

Thank you for submitting your article "Tissue damage drives co-localization of NF-κB, Smad3, and Nrf2 to direct Rev-erb sensitive wound repair in macrophages" for consideration by *eLife*. Your article has been reviewed by two reviewers including Joachim Schultze, and the evaluation was overseen by Satyajit Rath as the Reviewing Editor and Aviv Regev as the Senior Editor.

The reviewers have discussed the reviews with one another and the Reviewing Editor has drafted this decision to help you prepare a revised submission.

Summary:

The work is an outstanding example of understanding macrophage biology in context of complex environmental signals such as those present during wound healing. The authors show that macrophages integrate their signals in a context-specific fashion and that this is mainly driven by combinatorial action of several transcription factors, something that was also recently suggested for human monocyte-derived macrophages assessing reductionist in vitro models (Schmidt et al., Cell Research, 2016). This work is a logical and important next step towards understanding of the complex regulation and integration of environmental signals following reductionist models studied so far, and demonstrating that it is absolutely essential to overcome the previous simple model of polarized macrophage activation.

There are some concerns about both the data and their physiological significance and presentation as identified below, and enthusiasm for the manuscript will be increased if they are addressed. While all new physiological experiments mentioned below are not essential, it would be appropriate for the authors to provide some data in these areas and to acknowledge these issues.

Essential revisions:

1) Instead of starting with the by now already outdated M1/M2 model, it is suggested that the authors start with the more recent model of macrophage activation, the multi-dimensional model of macrophage activation (Ginhoux et el, Nature Immunology, 2016). It would be best not to use the terms M1/M2 at all; according to a new nomenclature suggested by Murray and colleagues (Immunity, 2014), terminology would be best changed throughout the manuscript (M1 = M(IFNγ), M2 = M(IL4)). This does not take away the major message of the story but would better reflect the current thought process in explaining how signals from the tissue and complex patterns of stress signals (as modeled here by tissue homogenates) are integrated by macrophages. In this context, the manuscript could introduce the reductionist stimuli (KLA, IL4, TGFβ) as a framework and could predict that combinatorial signals such as those from tissue homogenates would lead to the identification of so far unknown combinations of transcriptional regulators, mainly transcription factors associated with particular functions of macrophages, here response to wounding. This would provide a nice illustration of how one can use reductionist models and combine them into models with high relevance in vivo.

2) Several of the RNA-seq experiments were only done at n<3 replicates (as indicated in figure legends). It would be important to improve the dataset to at least n=3 for all conditions.

3) The mechanistic connections between the chromatin and transcription modification by and the functional outcome/s of Rev-erb deletion are somewhat thin. The bone marrow transfer-and-wound healing studies as they stand, while nice, do not show that it is indeed a macrophage-specific Rev-erb-deficiency that provides the accelerated wound healing phenotype since Tie2-Cre will likely mark all hematopoietic cell lineages in addition to endothelial lineage cells, nor do they rigorously establish the 'cell-autonomous' nature of the functional consequences of Rev-erb-deficiency for wound healing. Addressing these concerns definitively may require, for example, doing wound healing experiments in mixed bone marrow chimeras made with WT versus 'DKO' bone marrow mixed with macrophage-deficient strain bone marrow, and tracking relative recruitment of WT and 'DKO' macrophages (and neutrophils) post-wounding in mixed bone marrow chimeras made with WT plus 'DKO' bone marrow cells.

4) Secondly, the monocytic lineage landscape of the Rev-erb deficient mice is not characterized. This is relevant both in the context of specific macrophage 'subsets' that may contribute more or less to wound healing, and in the context of the specificity of the wound-healing functional phenotype vis-à-vis other macrophage functions. Thus, are 'patrolling' monocytes, a subset commonly claimed to contribute to wound healing, over-represented in 'DKO' cells? While the single-cell data suggest that this may not be the major explanation for the phenotype, a Rev-erb-deficiency-based modification of the differentiation-diversification of monocytic lineages would inevitably complicate the situation. Addressing this concern may require a detailed enumeration of the macrophage sub-lineages/subsets that the field currently recognizes. As for the specificity of the functional phenotype, some sense of whether (and the extent to which) other macrophage functions such as bacterial killing and cytokine/chemokine generation would help greatly.

5) Finally, since Rev-erb deficiency likely modifies the basal programs of deficient macrophages as the authors already note for some modules, distinguishing between 'prior' versus 'post-activation' roles of Rev-erb in macrophages is an essential discussion, and its elucidation may eventually well require use of inducible Cre-expression strategies (such as ERT2-Cre), and measurements of expression levels of specific target genes in WT versus 'DKO' macrophages in situ in infiltrated wounds.

6) The manuscript would further benefit from some more precision within the Results section.

---

## [Author Response]

*There are some concerns about both the data and their physiological significance and presentation as identified below, and enthusiasm for the manuscript will be increased if they are addressed. While all new physiological experiments mentioned below are not essential, it would be appropriate for the authors to provide some data in these areas and to acknowledge these issues.*

*Essential revisions:*

1) Instead of starting with the by now already outdated M1/M2 model, it is suggested that the authors start with the more recent model of macrophage activation, the multi-dimensional model of macrophage activation (Ginhoux et el, Nature Immunology, 2016). It would be best not to use the terms M1/M2 at all; according to a new nomenclature suggested by Murray and colleagues (Immunity, 2014), terminology would be best changed throughout the manuscript (M1 = M(IFNγ), M2 = M(IL4)). This does not take away the major message of the story but would better reflect the current thought process in explaining how signals from the tissue and complex patterns of stress signals (as modeled here by tissue homogenates) are integrated by macrophages. In this context, the manuscript could introduce the reductionist stimuli (KLA, IL4, TGFβ) as a framework and could predict that combinatorial signals such as those from tissue homogenates would lead to the identification of so far unknown combinations of transcriptional regulators, mainly transcription factors associated with particular functions of macrophages, here response to wounding. This would provide a nice illustration of how one can use reductionist models and combine them into models with high relevance in vivo.

We agree with the reviewers’ point of view on this and now begin the Introduction with the more recent multi-dimensional model of macrophage activation (Ginhoux et al., Nature Immunology, 2016) and incorporated the new nomenclature suggested by Murray and colleagues (Immunity, 2014). We also changed the text to introduce the reductionist stimuli as a framework that can be used to predict how combinations of transcriptional regulators coordinate immune and tissue repair activities in complex tissue microenvironments.

2) Several of the RNA-seq experiments were only done at n<3 replicates (as indicated in figure legends). It would be important to improve the dataset to at least n=3 for all conditions.

To address this concern, we performed additional experiments so that all RNA-Seq data sets are now at least n=3 with the exception of IL4 treatments (n=2), which we did not expand because this treatment turned out to be a relatively minor aspect of the manuscript. All of the heatmaps and corresponding analyses have been revised to incorporate these new experiments (Figure 1, Figure 1—figure supplement 1, Figure 3, Figure 3, Figure 6).

3) The mechanistic connections between the chromatin and transcription modification by and the functional outcome/s of Rev-erb deletion are somewhat thin. The bone marrow transfer-and-wound healing studies as they stand, while nice, do not show that it is indeed a macrophage-specific Rev-erb-deficiency that provides the accelerated wound healing phenotype since Tie2-Cre will likely mark all hematopoietic cell lineages in addition to endothelial lineage cells, nor do they rigorously establish the 'cell-autonomous' nature of the functional consequences of Rev-erb-deficiency for wound healing. Addressing these concerns definitively may require, for example, doing wound healing experiments in mixed bone marrow chimeras made with WT versus 'DKO' bone marrow mixed with macrophage-deficient strain bone marrow, and tracking relative recruitment of WT and 'DKO' macrophages (and neutrophils) post-wounding in mixed bone marrow chimeras made with WT plus 'DKO' bone marrow cells.

We agree that these experiments do not show that it is a macrophage-specific Rev-erb deficiency that accounts for the accelerated wound healing phenotype. We gave serious consideration to the suggestion to perform mixed chimera experiments, and as described below, performed new bone marrow transplantation experiments to address other concerns. However, we were not able to devise an experimental plan using existing strains of mice that would provide a clear-cut answer to this question. The most straightforward approach would be to use alternative deleter strains to generate mice in which the Rev-erbs are selectively and/or inducibly deleted in macrophages as suggested in Point 5, below. However, even these approaches are not definitive, e.g., the conventional LysM-Cre strain also deletes genes in granulocytes, which could contribute to the wound healing phenotype, and does not delete efficiently in some of the resident macrophage populations of the skin. Our main purpose in presenting the wound phenotype was to provide a biological context for the subsequent molecular analyses of the effects of complex signals on transcription factor interactions. While we think that the wound repair findings are significant in themselves and will be of general interest, restricting or expanding the cell types involved would not alter the main conclusions of the manuscript. We therefore hope that a definitive answer to this question will not be required for publication in *eLife*. To address the Reviewer’s concerns on this point, we more clearly state in the Results and Discussion that the accelerated wound repair phenotype results from a loss of Rev-erb expression in cells derived from the bone marrow compartment and that further studies will be required to establish the relative contributions of macrophages and other bone marrow-derived cells.

4) Secondly, the monocytic lineage landscape of the Rev-erb deficient mice is not characterized. This is relevant both in the context of specific macrophage 'subsets' that may contribute more or less to wound healing, and in the context of the specificity of the wound-healing functional phenotype vis-à-vis other macrophage functions. Thus, are 'patrolling' monocytes, a subset commonly claimed to contribute to wound healing, over-represented in 'DKO' cells? While the single-cell data suggest that this may not be the major explanation for the phenotype, a Rev-erb-deficiency-based modification of the differentiation-diversification of monocytic lineages would inevitably complicate the situation. Addressing this concern may require a detailed enumeration of the macrophage sub-lineages/subsets that the field currently recognizes. As for the specificity of the functional phenotype, some sense of whether (and the extent to which) other macrophage functions such as bacterial killing and cytokine/chemokine generation would help greatly.

We thank the reviewers for the important suggestion to exclude possible changes in monocyte subsets. To address this concern, we performed a new round of bone marrow transplantation using WT or DKO bone marrow to enable an extensive flow cytometry analysis of each genotype in the same background. Based on these analyses, there was no difference in the ratio of specific macrophage subsets (Ly6c^high^ or Ly6c^low^) or neutrophils (neutrophils data not shown, p-value=0.5287 Welch two sample t-test) in peripheral blood WT and Rev-erb DKO bone marrow transplanted animals (please see Figure 2—figure supplement 1). With regards to cell autonomous functional phenotypes, we demonstrate that Rev-erb DKO macrophages exhibit enhanced migration through matrigel (Figure 2), which was predicted from the upregulation of Mmp9.

5) Finally, since Rev-erb deficiency likely modifies the basal programs of deficient macrophages as the authors already note for some modules, distinguishing between 'prior' versus 'post-activation' roles of Rev-erb in macrophages is an essential discussion, and its elucidation may eventually well require use of inducible Cre-expression strategies (such as ERT2-Cre), and measurements of expression levels of specific target genes in WT versus 'DKO' macrophages in situ in infiltrated wounds.

We agree with the reviewers that Rev-erb deficiency likely modifies both basal and signal dependent transcriptional programs, and that to distinguish between “prior” versus “post-activation” roles of Rev-erbs in macrophages may require the use of inducible Cre-expression strategies, as well as measurements of target gene expression in situ in macrophage infiltrated wounds. This essential discussion point has now been incorporated in the manuscript.